# S$^2$M-Former: Spiking Symmetric Mixing Branchformer for Brain Auditory Attention Detection

**Jiaqi Wang**[1,2], **Zhengyu Ma**[2]\*, **Xiongri Shen**[1], **Chenlin Zhou**[2,3], **Leilei Zhao**[1], **Han Zhang**[2,4]
**Yi Zhong**[1,5], **Siqi Cai**[1,6], **Zhenxi Song**[1], **Zhiguo Zhang**[1,6]\*

[1]Harbin Institute of Technology, Shenzhen    [2]Pengcheng Laboratory    [3]Peking University
[4]Harbin Institute of Technology    [5]Great Bay University    [6]Shenzhen Loop Area Institute
{mhwjq1998}@gmail.com; {mazhy}@pcl.ac.cn; {zhiguozhang}@hit.edu.cn

## Abstract

Auditory attention detection (AAD) aims to decode listeners' focus in complex auditory environments from electroencephalography (EEG) recordings, which is crucial for developing neuro-steered hearing devices. Despite recent advancements, EEG-based AAD remains hindered by the absence of synergistic frameworks that can fully leverage complementary EEG features under energy-efficiency constraints. We propose *S$^2$M-Former*, a novel *s*piking *s*ymmetric *m*ixing framework to address this limitation through two key innovations: i) Presenting a spike-driven symmetric architecture composed of parallel spatial and frequency branches with mirrored modular design, leveraging biologically plausible token-channel mixers to enhance complementary learning across branches; ii) Introducing lightweight 1D token sequences to replace conventional 3D operations, reducing parameters by 14.7×. The brain-inspired spiking architecture further reduces power consumption, achieving a 5.8× energy reduction compared to recent ANN methods, while also surpassing existing SNN baselines in terms of parameter efficiency and performance. Comprehensive experiments on three AAD benchmarks (KUL, DTU and AV-GC-AAD) across three settings (within-trial, cross-trial and cross-subject) demonstrate that S$^2$M-Former achieves comparable state-of-the-art (SOTA) decoding accuracy, making it a promising low-power, high-performance solution for AAD tasks. Code is available at `https://github.com/JackieWang9811/S2M-Former`.

## 1 Introduction

The "cocktail party effect" refers to the remarkable ability of the human auditory system to isolate and focus on a specific speaker's speech in a competitive background and noise environment [1, 2]. This capacity, realized through dynamic neural processing in the auditory cortex and top-down attentional modulation [3], has profound implications for understanding human auditory cognition and developing neuroengineering applications. Auditory attention detection (AAD) investigates the brain's selective hearing ability by detecting which sound stream a listener is focusing on, based on neural recordings, as illustrated in Figure. 1. It aims to address a critical challenge in neural rehabilitation: ***How to restore natural auditory scene analysis for individuals with hearing impairments***.

Recent advances [4, 5, 6, 7] in non-invasive electroencephalography (EEG)-based approaches have demonstrated remarkable success in reconstructing attentional selection patterns from cortical responses. On the one hand, AAD enables hearing aids to dynamically amplify the speech stream that the user is focusing on, providing more natural and adaptive listening. On the other hand, integrating

---

\*Corresponding author

39th Conference on Neural Information Processing Systems (NeurIPS 2025).

AAD into brain-computer interface (BCI) systems allows real-time feedback between the brain and auditory devices, mimicking thalamocortical feedback pathways observed in natural auditory processing [8]. As AAD is increasingly integrated into wearable systems such as hearing aids [9, 10] and low-power BCIs [11], the need for models that are both accurate and energy-efficient becomes critical, due to strict constraints on battery life, latency, and computational resources. These practical demands highlight the importance of lightweight AAD models for real-world deployment.

Feature extraction techniques such as common spatial pattern (CSP) [12, 13] and differential entropy (DE) [14, 15] have demonstrated effectiveness in capturing discriminative characteristics from auditory-evoked EEG signals. Despite recent progress, AAD remains hindered by the absence of synergistic frameworks that can fully leverage complementary EEG features under energy-efficiency constraints. This limitation is particularly critical for real-world AAD applications, which require lightweight, low-power solutions for high-performance, closed-loop systems in hearing aids.

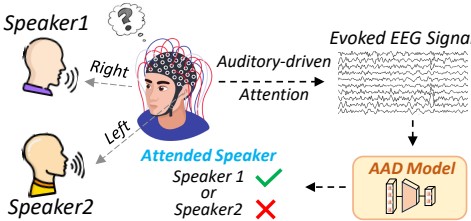

Figure 1: EEG-based AAD Paradigm.

Although recent dual-branch networks [6, 7] that incorporate multiple EEG features have shown improved performance over single-branch models, they still face several challenges. First, most of these methods adopt an isolated learning paradigm, combining features via simple concatenation or summation, neglecting the potential hypothesis that the complementary learning among EEG features can benefit performance [16, 17, 18]. Second, their designs introduce substantial computational overhead, especially when modeling specific EEG properties (e.g., topological structures) using resource-intensive operations such as 3D convolutions, thereby limiting deployability.

To tackle the above challenges, we introduce S$^2$M-Former, a spike-driven symmetric model that is naturally endowed with low power consumption by replacing energy-intensive multiply-accumulate (MAC) operations with sparse, spike-based accumulate (AC) communication, while retaining the precise temporal dynamics essential for modeling auditory attention. Specifically, the proposed S$^2$M-Former brings several key advantages. Firstly, the symmetric design enables parallel extraction of spatial and frequency-domain representations using mirrored modular structures. This design encourages complementary learning across branches by facilitating the synergistic fusion of domain-specific representations, thereby replacing isolated learning. Within each branch, multi-level modules capture hierarchical contextual dependencies, enabling expressive and informative representations without the need for complex customizations. Secondly, to address the high parameter count and computational complexity associated with dual-branch bringing, S$^2$M-Former embraces a lightweight, efficient architecture. By replacing power-hungry 3D operations with streamlined 1D token representations, we drastically cut down on computational overhead, achieving superior computational efficiency while preserving high performance. The above makes S$^2$M-Former particularly suited for energy-efficient neuromorphic neuro-steered devices.

We validate our model on three public AAD datasets: two uni-stimuli (audio-only) benchmarks, KUL [19] and DTU [20], and one multi-stimuli (audio-visual) benchmark, AV-GC-AAD [21], under within-trial, cross-trial and cross-subject settings. Compared to recent dual-branch models [6, 7], S$^2$M-Former reduces the parameter count by 14.7 times and power consumption by 5.8 times, all while achieving competitive SOTA performance. In comparison to the latest single-branch SOTA models [5, 15], S$^2$M-Former not only maintains fewer parameters but also roundly surpasses them. Furthermore, it achieves superior performance against recent influential SNN backbones. These results collectively underscore the effectiveness and efficiency of our proposed S$^2$M-Former in tackling the challenges posed by AAD tasks. Our contributions can be summarized as follows:

**1)** We propose S$^2$M-Former, the first spiking symmetric mixing framework for auditory attention decoding tasks, which enables effective hierarchical integration of contextual representations within each branch. By leveraging biologically plausible mixers, our design naturally promotes complementary learning across EEG feature branches, thereby notably improving performance.

**2)** S$^2$M-Former reduces the parameter count by up to $14.7\times$ and energy consumption by $5.8\times$ compared to recent dual-branch models, without requiring any complex customization. Its spike-driven architecture fully delivers higher decoding accuracy than both its ANN counterpart and existing SNN baselines, highlighting a compelling trade-off between efficiency and performance.

**3)** We validate S$^2$M-Former across three AAD datasets and multiple evaluation settings, where it achieves competitive state-of-the-art performance. These results demonstrate its strong generalization, offering a new perspective in low-power brain-computer interfaces for AAD tasks.

## 2 Related Works

**Brain Auditory Attention Detection.** In early development, CSP-CNN [12] combined CSP with CNNs to enhance the non-linear modeling capacity, demonstrating the potential for robust AAD. Following this, SSF-CNN [14] focused on the topographic distribution of alpha-band EEG power and introduced a spectro-spatial DE extraction strategy to decode auditory attention. MBSSFCC [15] extended the framework by incorporating multi-band frequency analysis, extracting DE features, and applying a ConvLSTM module for spectro-spatial-temporal learning. Recently, DBPNet [6] proposes a dual-branch network for AAD, which consists of a temporal attentive branch and a frequency 3D convolutional residual branch, and fuses the temporal-frequency domain features by concatenating the outputs from branches into a fusion vector. M-DBPNet [7] is an upgraded version of DBPNet. This framework introduces Mamba-based [22] methods in the temporal branch, aiming to better extract temporal features from sequential embeddings, with a few additional parameters. DARNet [5] is a dual attention refinement network designed to capture spatio-temporal representations and long-range dependencies in the CSP feature patterns, achieving near dual-branch performance.

**Spiking Neural Networks.** Recognized as the third generation of neural networks [23], spiking neural networks (SNNs) effectively mimic the dynamics of biological neurons with sparse and asynchronous spikes [24]. This approach enables SNNs to achieve high computational performance with low energy consumption, making them a viable energy-efficient alternative to artificial neural networks (ANNs) [25]. Despite these advancements, recent SNN methods for AAD [26, 27, 28] still showcase weaker performance compared to recent SOTA ANN models, primarily due to their sparse feature representation and relatively simple network architectures, and face challenges such as closed-source implementations and limited reproducibility, making direct comparisons difficult.

## 3 S$^2$M-Former

As illustrated in Figure. 2, S$^2$M-Former is an SNN model that leverages membrane potential-based transmission [29] with shortcut connections to maintain spike-driven dynamics across layers. Given an EEG series $E$, we first extract spatial-temporal features ($E_S \in \mathbb{R}^{C \times T}$, with $C$ denoting channels and $T$ time points) using CSP and frequency-spectral features ($E_F \in \mathbb{R}^{5 \times H \times W}$, where $H \times W$ is the map size) via DE, as detailed in Appendix A.1. These embeddings are then expanded along the temporal dimension over $T_S$ steps and encoded by branch-specific spiking encoders. The representations are further refined through a series of spike-driven modules within the spiking symmetric mixing (S$^2$M) block, effectively capturing complementary spatial-frequency patterns. Finally, the fused $D$-dimensional embeddings are passed through a classification head to produce the prediction $\hat{Y}$.

### 3.1 Spiking Neuron

Spiking neurons serve as bio-plausible abstractions of neural activity [30, 31]. We adopt the Leaky Integrate-and-Fire (LIF) [32] neuron for *intra-module communication*, whose discrete-time dynamics are defined in Eq. (20), Eq. (21) and Eq. (22). To enhance membrane potential awareness across *inter-module connections* within the S$^2$M-Former, we propose a novel neuron variant: the channel-wise parametric LIF (CPLIF) neuron. It builds upon the parametric LIF neuron—a variant of LIF with a learnable membrane time constant that enables adaptive temporal control [33]—by assigning an individual time constant to each channel. This design enables channel-wise adaptive modeling of temporal dynamics, allowing more expressive and finer-grained spiking activation across time steps. We denote the CPLIF as $\mathcal{SN}_{head}$ in subsequent sections. Its membrane potential update is given by:

$$H[t, c, n] = V[t-1, c, n] + \frac{1}{\tau_l[c]} \left( X[t, c, n] - (V[t-1, c, n] - V_{reset}) \right) + \beta[c], \quad (1)$$

where $\tau_l, \beta \in \mathbb{R}^C$ are learnable vectors for the channel-wise membrane time constants and bias, respectively. $c$ indexes the channel dimension and $n$ denotes the token index within each channel. The CPLIF shares the same firing and reset equations as the LIF, with details provided in Appendix A.2.

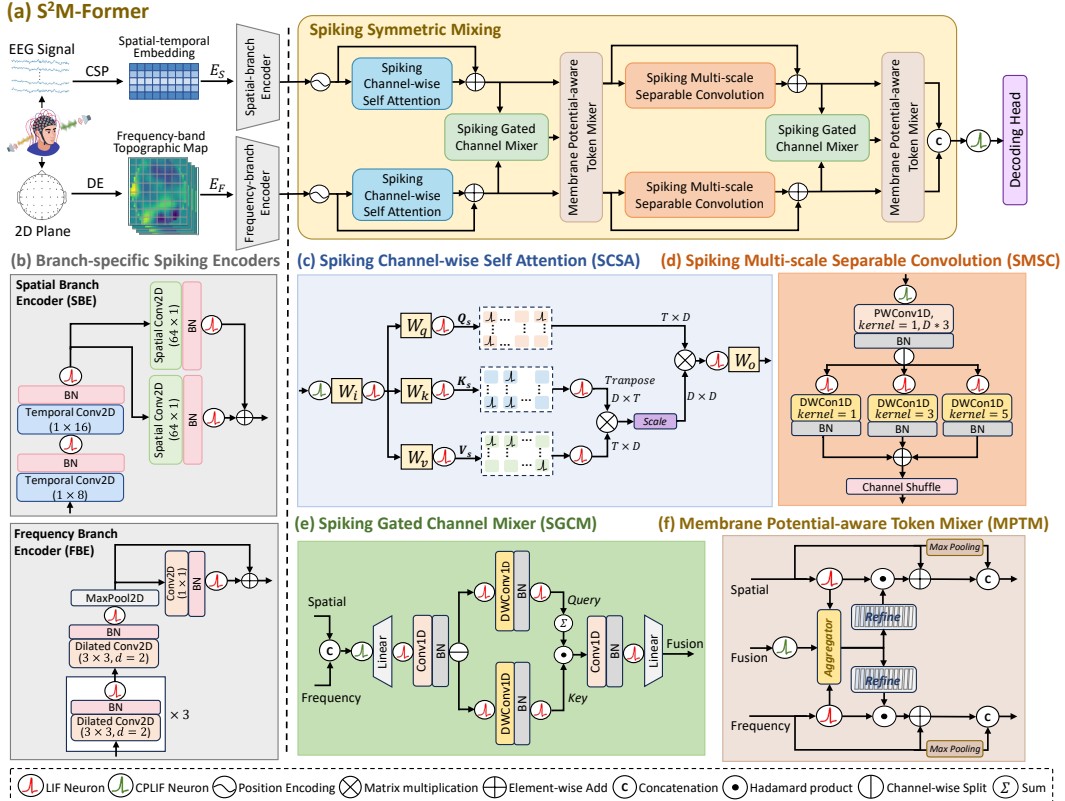

Figure 2: S²M-Former: A dual-branch mirrored architecture comprising branch-specific spiking encoders for diverse domain features, leveraging spike-driven symmetric modules for contextual representation learning and enabling effective complementary interactions across parallel branches.

## 3.2 Branch-specific Spiking Encoders

**Spatial Branch Encoder (SBE).** To effectively capture spatio-temporal EEG dynamics [5], we design a spike-driven SBE that jointly models temporal dependencies and spatial correlations across electrodes. The SBE comprises two key components: (1) biologically plausible spiking neuron dynamics to enable spike-driven representation learning; and (2) a multi-stage mechanism that integrates progressive condensation with refined integration. Given the input feature map $E_S \in \mathbb{R}^{T_S \times C \times T}$, we first apply a cascade of temporal convolution layers ($TConv2d$) with increasing kernel sizes (e.g., $k = 8$) to progressively extract temporal dependencies. Each temporal layer is followed by batch normalization ($BN$) and a spiking neuron $\mathcal{SN}(\cdot)$. The operations are formally described as:

$$E'_S = \mathcal{SN}(BN(TConv2d_{(1 \times k)}(E_S))) \in \mathbb{R}^{T_S \times 2D \times C \times T}, \tag{2}$$

$$E''_S = \mathcal{SN}(BN(TConv2d_{(1 \times 2k)}(E'_S))) \in \mathbb{R}^{T_S \times 4D \times C \times T}. \tag{3}$$

Next, we introduce a dual-path spatial convolution module ($SConv2d$), each path employing a full-channel spatial filter with kernel size ($C \times 1$), which aggregates cross-channel interactions. The outputs of both spatial paths are then refined via residual addition to enhance the spatial representation:

$$E_S^{out} = \mathcal{SN}(BN(SConv2d_{(C \times 1)}(E''_S))) + \mathcal{SN}(BN(SConv2d_{(C \times 1)}(E''_S))) \in \mathbb{R}^{T_S \times D \times T}. \tag{4}$$

**Frequency Branch Encoder (FBE).** Recent spiking patch splitting modules [34, 35] are primarily designed for visual inputs. To effectively extract frequency-spectral-spatial representations from EEG 2D brain topography, we propose a novel FBE, which addresses both the frequency-specific information and region-specific activation patterns inherent in multi-band EEG topographic data. FBE begins by applying three successive spiking convolutional operations:

$$E'_F = \mathcal{SN}(\text{BN}(\text{Conv2d}(E_{F_l}))) \in \mathbb{R}^{T_S \times D_l \times H \times W}, \quad l \in \{1, \dots, L\}, \tag{5}$$

where $E_{F_l} \in \mathbb{R}^{T_S \times D_l \times H \times W}$ is the input at the $l$-th block ($L = 3$), and each $3 \times 3$ convolution employs a dilation rate of 2 to expand the receptive field. This enables the model to capture spatially dispersed yet salient edge-region activations more effectively [36]. In addition, the channel dimensions are progressively transformed via $D \rightarrow 4D \rightarrow 2D \rightarrow D$, followed by a max-pooling layer for spatial compression. To further refine the representation, we apply a $1 \times 1$ spiking convolutional layer with a residual connection, enhancing representation stability and robustness.

## 3.3  S²M Module

After processing through the spatial (SBE) and frequency (FBE) encoders, EEG embeddings are unified as 1D tokens $X \in \mathbb{R}^{T_S \times N \times D}$, where $T_S$ denotes the number of time steps, $N$ is the number of tokens, and $D$ is the feature dimension. As shown in Figure. 2 (a), the S²M block hierarchically refines these representations through a series of mirrored spike-driven modules: the SCSA module first captures long-range intra-branch dependencies, followed by SGCM and MPTM for cross-branch fusion across channel and token dimensions. SMSC then extracts local fine-grained details, and a second combination of SGCM and MPTM further reinforces symmetric complementary integration.

### 3.3.1  Spiking Channel-wise Self Attention (SCSA)

Recent spiking attention mechanisms have made considerable progress, with spiking self-attention (SSA) being one of the most widely applied methods [34]. In our implementation, we modify the SSA mechanism by adopting a channel-wise protocol to enhance interpretability, where the spatial branch reveals electrode correlations, and the frequency branch uncovers multi-band relationships. Moreover, this adjustment reduces the spiking attention score matrix from $N \times N$ to $D \times D$ [37], leading to a lower computational complexity from $O(N^2 D)$ to $O(ND^2)$ for greater efficiency. As illustrated in Figure. 2 (c), we define this module as SCSA. Specifically, given an input EEG embedding $X \in \mathbb{R}^{T_S \times N \times D}$, the computation proceeds as follows:

$$X_S = \mathcal{SN}_I(\boldsymbol{W}_I(\mathcal{SN}_{head}(X))) \in \mathbb{R}^{T_S \times N \times D}, \tag{6}$$

$$Q_S = \mathcal{SN}_Q(\boldsymbol{W}_Q(X_S)), \ K_S = \mathcal{SN}_K(\boldsymbol{W}_K(X_S)), V_S = \mathcal{SN}_V(\boldsymbol{W}_V(X_S)), \tag{7}$$

$$U = \mathcal{SN}\left((K_S^T V_S)Q_S * \alpha\right) \in \mathbb{R}^{T_S \times N \times D}, \tag{8}$$

$$X_{out} = \mathcal{SN}_O(\boldsymbol{W}_O(U)) \in \mathbb{R}^{T_S \times N \times D}, \tag{9}$$

where $Q_S$, $K_S$, and $V_S$ are the spiking query, key, and value vectors, each of size $\mathbb{R}^{T_S \times N \times D}$, $\alpha$ denotes a scale factor. Each learnable projection matrix $\boldsymbol{W}$ is implemented as a 3 kernel size of 1D depth-wise convolution, allowing spatially local parameter-efficient feature transformation.

### 3.3.2  Spiking Multi-scale Separable Convolution (SMSC)

We propose the SMSC module, which leverages multiple depth-wise convolutions with different receptive fields to enrich multi-scale patterns in EEG embeddings. As shown in Figure. 2(d), the module comprises three parallel depthwise convolutional paths with different receptive fields, followed by a channel shuffle operation to promote cross-scale and cross-channel information integration. Given the input spiking feature $X \in \mathbb{R}^{T_S \times N \times D}$, we first apply a CPLIF neuron $\mathcal{SN}_{head}$, followed by pointwise convolution ($PWConv1d$) and batch normalization:

$$X' = BN(PWConv1d(\mathcal{SN}_{head}(X))) \in \mathbb{R}^{T_S \times N \times 3D}. \tag{10}$$

Here, PWConv expands the channel dimension for later multi-scale processing. The feature $X'$ is evenly split into three parts along the channel dimension. Each part is then processed by a depthwise convolution ($DWConv1d_{k_i}$), where a distinct kernel size $k_i \in \{1, 3, 5\}$ is assigned to extract localized patterns at different temporal scales:

$$X''_i = BN_i(DWConv1d_{k_i}(\mathcal{SN}_i(\text{Split}(X', 3)))) \in \mathbb{R}^{T_S \times N \times D}; \quad i \in \{1, 2, 3\}. \tag{11}$$

Unlike standard convolutions that aggregate information across channels, depthwise convolution processes each channel independently, thereby reducing computation but limiting inter-channel interaction. To compensate for this, we adopt a parameter-free channel shuffle operation [38] after summing the three branches:

$$X''' = \text{Shuffle}(X''_1 + X''_2 + X''_3). \tag{12}$$

This operation rearranges the channel dimensions by grouping and permuting features to facilitate cross-group feature integration, thereby enhancing representation diversity without adding complexity.

### 3.3.3 Spiking Gated Channel Mixer (SGCM)

The SGCM module is designed to adaptively fuse multi-channel spatial and frequency representations. Firstly, $X_S$ and $X_F$ are concatenated which can form as $X' = \text{Concat}(X_S, X_F) \in \mathbb{R}^{T_S \times (N_S + N_F) \times D}$, where $N_S$ and $N_F$ are the number of brach-specific tokens, respectively, and $D$ is the feature dimension. The concatenated features $X'$ are fed into a spiking neuron followed by a linear transformation, whose channel dimensions are projected from $D \rightarrow 2D$, then passed through a convolution stem to refine the representations:

$$X'' = BN(Conv1d(\mathcal{SN}(Linear(\mathcal{SN}_{head}(X'))))) \in \mathbb{R}^{T_S \times (N_S + N_F) \times 2D}. \tag{13}$$

As presented in Figure. 2 (e), the core of the SGCM integrates a spiking gating mechanism, modulating the flow of information across channels. Specifically, the $X''$ is split into two components,

$$X_Q, X_K = \mathcal{SN}_{q_2, k_2}(BN_{q,k}(DWConv_{q,k}(\mathcal{SN}_{q_1, k_1}(\text{Split}(X'', 2))))) \in \mathbb{R}^{T_S \times (N_S + N_F) \times D}, \tag{14}$$

where $X_Q$ and $X_K$ sever as the query and key value [39], respectively, and the calculation process of the gating mechanism can be formulated as: $A_c = \mathcal{SN}(\sum_{i=0}^{D} X_{Q_{i,j}}) \in \mathbb{R}^{T_S \times (N_S + N_F) \times 1}$, where $A_c$ is the channel attention vector, which models the importance of different channels. Then the channel-wise mask is adopted by $\mathbf{X}''' = A_c \odot X_K$. Finally, the attention-modulated gated embeddings are calculated to generate the final output:

$$X_{fusion} = Linear(\mathcal{SN}_{out}(BN(Conv1d(\mathbf{X}''')))) \in \mathbb{R}^{T_S \times (N_S + N_F) \times D}. \tag{15}$$

### 3.3.4 Membrane Potential-aware Token Mixer (MPTM)

As illustrated in Figure. 2 (f), the MPTM is primarily structured as two parallel operations: the fusion branch from SGCM modulates the frequency and spatial branches, respectively. For clarity, we detail the interaction between the fusion branch $X_{fusion}$ and a generic branch $X_G \in \mathbb{R}^{T_S \times N \times D}$, which denotes either the spatial or frequency branch, and decompose the MPTM into three functional stages.

**Core Representation.** We implement a global average pooling (GAP) as an *Aggregator* over the token dimension to encode global information across all tokens and thereby capture long-range dependencies [40]. This operation condenses the spiking membrane potentials into a compact global summary that serves as the core representation for cross-branch fusion, formulated as:

$$X'_G = GAP(\mathcal{SN}(X_G)) \in \mathbb{R}^{T_S \times 1 \times D}, X'_{fusion} = GAP(\mathcal{SN}_{head}(X_{fusion})) \in \mathbb{R}^{T_S \times 1 \times D}. \tag{16}$$

**Representation Refinement and Fusion.** To facilitate fine-grained potential-aware modulation using the core representation, we construct a refined guidance representation, denoted as $R \in \mathbb{R}^{T_S \times N \times D}$, by proportionally combining the global summaries from two branches. Specifically, we divide the total number of tokens $N$ into two parts: $N_G = \lfloor \alpha N \rfloor$ represents the number of tokens filled from the generic branch, and $N_{fusion} = N - N_G$ tokens are filled by repeating from the fusion branch, where we set $\alpha = 0.5$. We then repeat and concatenate them, which can be written as:

$$R = \text{Concat}(\text{Repeat}(X'_G, N_G), \text{Repeat}(X'_{fusion}, N_{fusion})) \in \mathbb{R}^{T_S \times N \times D}. \tag{17}$$

The final fused output is denoted as $F$, which integrates the refined guidance $R$ and the original primary features through a spiking element-wise modulation with residual connection, formulated as:

$$F = \mathcal{SN}(X_G) \odot R + X_G \in \mathbb{R}^{T_S \times N \times D}. \tag{18}$$

**Output Generation.** The final output is generated by concatenating the fused representation with the compressed original input $X_G$, where max pooling is applied with a kernel size of 3 and a stride of 2.

$$X_{out} = \text{Concat}(F, MaxPool(X_G)) \in \mathbb{R}^{T_S \times \frac{3}{2} N \times D}. \tag{19}$$

## 4 Experiments

**Datasets and Processing.** We evaluate the performance of S$^2$M-Former across three publicly available AAD datasets: KUL and DTU, which focus on auditory-only stimuli, and AV-GC-AAD, which includes audio-visual stimuli. A summary of the datasets is provided in Table 1. Each dataset is preprocessed using pipelines aligned with prior AAD studies to ensure consistency across

Table 1: Comprehensive statistics and details for the three AAD datasets.

| Dataset | Subjects | Scene | Language | Trials | Duration per trial (seconds) | Duration per subject (minutes) | Total duration (hours) |
|---|---|---|---|---|---|---|---|
| KUL [19] | 16 | audio-only | Dutch | 8 | 360 | 48 | 12.8 |
| DTU [20] | 18 | audio-only | Danish | 60 | 50 | 50 | 15.0 |
| AV-GC [21] | 11 | audio-visual | Dutch | 8 | 600 | 80 | 14.7 |

methods: The KUL dataset [19] is re-referenced to the central electrode, bandpass filtered (0.1-50 Hz), downsampled to 128 Hz, and trimmed to the same trial length. The DTU dataset [20] is high-pass filtered at 0.1 Hz, downsampled to 128 Hz, and denoised at 50 Hz. Eye artifacts are corrected via joint decorrelation, followed by re-referencing to the channel average. The AV-GC dataset [21] is bandpass filtered (1-40 Hz), re-referenced to the average of all channels, and downsampled to 128 Hz [41]. To avoid contaminating preprocessing signatures that could affect CSP filters [21], no z-scoring or time normalization is applied to any of the datasets. More implementation details are provided in Appendix A.3 and A.4.

**Evaluation Methods.** The subjects in the three public AAD datasets were instructed to focus on one of two simultaneous speakers, with the directions being left and right, formulating it as a classification task with the binary labels. We evaluate model performance using the average accuracy and standard deviation (SD) across all subjects for each experiment and decision window. We reproduce and compare five publicly available baseline models (details in Section 2): **SSF-CNN** [14], **MBSSFCC** [15], **DARNet** [5], **DBPNet** [6], and **M-DBPNet** [7] under two subject-dependent evaluations and a subject-independent evaluation:

- **Within-trial** [5, 6, 15]: EEG data from a single trial are split into 8:1:1 train, validation and test sets. The final train/validation/test sets are obtained by concatenating data across all trials.
- **Cross-trial** [7, 42, 43]: To prevent overfitting to specific EEG segments, all trials are randomly split into 8:1:1 for training, validation, and testing. For trials with fewer than 10 samples, two with different labels are randomly chosen for testing, and the rest are proportionally divided.
- **Cross-subject** [43]: We follow a leave-one-subject-out (LOSO), where leaving out one subject's data as the test set while training on the remaining subjects, iteratively performing cross-validation.

To address real-time response needs and ensure fair benchmarking, all models are evaluated under short decision windows of 0.1s, 1s, and 2s. All methods (including baselines) adopt the same preprocessing pipeline, followed by EEG segmentation using a sliding window with 50% overlap, in line with standard practice in prior AAD works. Feature extraction is performed separately per set to prevent information leakage.

Table 2: Within-trial results under three datasets. Color shading: ■Highest, ■Second, ■Third.

| Datasets | Models | Architecture | Params (M) | Accuracy (%) ± SD | | |
|---|---|---|---|---|---|---|
| | | | | 2-second | 1-second | 0.1-second |
| KUL | SSF-CNN [14] | Single | 4.21 | 79.64 ± 9.64 | 76.63 ± 10.28 | 77.73 ± 9.60 |
| | MBSSFCC [15] | | 16.81 | 93.71 ± 6.46 | 92.65 ± 7.48 | 84.02 ± 9.39 |
| | DARNet [5] | | 0.08 | 92.81 ± 9.45 | 92.04 ± 9.75 | 87.66 ± 10.79 |
| | DBPNet [6] | Dual | 0.88 | 93.66 ± 7.88 | 93.25 ± 7.33 | 85.70 ± 9.75 |
| | M-DBPNet [7] | | 1.32 / 1.00 / 0.88 | 93.75 ± 6.34 | 93.19 ± 7.28 | 86.16 ± 9.94 |
| | S²M-Former (ours) | | **0.06** | 93.71 ± 8.14 | 92.27 ± 8.66 | 83.39 ± 12.80 |
| DTU | SSF-CNN [14] | Single | 4.21 | 70.65 ± 6.18 | 67.63 ± 4.35 | 65.44 ± 4.72 |
| | MBSSFCC [15] | | 16.81 | 80.20 ± 7.62 | 76.64 ± 7.97 | 69.43 ± 5.59 |
| | DARNet [5] | | 0.08 | 81.30 ± 5.76 | 79.89 ± 7.88 | 76.04 ± 6.60 |
| | DBPNet [6] | Dual | 0.88 | 83.93 ± 5.17 | 80.69 ± 6.54 | 77.06 ± 5.08 |
| | M-DBPNet [7] | | 1.32 / 1.00 / 0.88 | 82.56 ± 8.01 | 81.12 ± 6.82 | 74.06 ± 5.68 |
| | S²M-Former (ours) | | **0.06** | 85.28 ± 6.01 | 82.87 ± 6.92 | 75.84 ± 5.46 |
| AV-GC | SSF-CNN [14] | Single | 4.21 | 79.50 ± 8.45 | 76.40 ± 7.96 | 66.67 ± 5.37 |
| | MBSSFCC [15] | | 16.81 | 89.13 ± 7.21 | 87.90 ± 6.98 | 72.17 ± 5.77 |
| | DARNet [5] | | 0.08 | 89.17 ± 6.94 | 88.31 ± 6.87 | 79.45 ± 7.70 |
| | DBPNet [6] | Dual | 0.88 | 90.78 ± 4.91 | 89.04 ± 5.15 | 73.37 ± 6.13 |
| | M-DBPNet [7] | | 1.32 / 1.00 / 0.88 | 87.04 ± 7.76 | 86.26 ± 7.35 | 72.88 ± 6.73 |
| | S²M-Former (ours) | | **0.06** | 91.83 ± 6.66 | 89.24 ± 7.59 | 74.42 ± 7.79 |

Table 3: Cross-trial results under three datasets with color shading: ■ Highest, ■ Second, ■ Third.

| Datasets | Models | Architecture | Params (M) | Accuracy (%) ± SD | | |
|---|---|---|---|---|---|---|
| | | | | 2-second | 1-second | 0.1-second |
| KUL | SSF-CNN [14] | Single | 4.21 | 60.32 ± 19.25 | 58.59 ± 17.27 | 61.93 ± 15.88 |
| | MBSSFCC [15] | | 16.81 | 73.56 ± 23.98 | 71.01 ± 22.50 | 65.12 ± 20.63 |
| | DARNet [5] | | 0.08 | 68.92 ± 24.06 | 68.43 ± 24.18 | 68.01 ± 22.63 |
| | DBPNet [6] | Dual | 0.88 | 72.95 ± 24.36 | 70.89 ± 25.01 | 65.67 ± 21.84 |
| | M-DBPNet [7] | | 1.32 / 1.00 / 0.88 | 74.27 ± 21.37 | 70.64 ± 23.82 | 66.97 ± 21.87 |
| | S²M-Former (ours) | | **0.06** | 72.39 ± 25.21 | 71.22 ± 25.97 | 66.49 ± 21.03 |
| DTU | SSF-CNN [14] | Single | 4.21 | 69.50 ± 7.28 | 67.25 ± 7.44 | 65.21 ± 9.63 |
| | MBSSFCC [15] | | 16.81 | 76.53 ± 8.84 | 75.55 ± 8.86 | 70.08 ± 8.36 |
| | DARNet [5] | | 0.08 | 72.41 ± 9.73 | 71.98 ± 10.03 | 69.11 ± 8.67 |
| | DBPNet [6] | Dual | 0.88 | 76.40 ± 9.53 | 74.25 ± 10.03 | 66.54 ± 6.92 |
| | M-DBPNet [7] | | 1.32 / 1.00 / 0.88 | 76.18 ± 9.18 | 74.86 ± 9.50 | 67.57 ± 8.12 |
| | S²M-Former (ours) | | **0.06** | 76.74 ± 9.96 | 75.75 ± 9.96 | 70.36 ± 7.31 |
| AV-GC | SSF-CNN [14] | Single | 4.21 | 64.42 ± 11.14 | 63.45 ± 9.87 | 59.62 ± 7.79 |
| | MBSSFCC [15] | | 16.81 | 63.67 ± 16.48 | 60.82 ± 13.82 | 60.19 ± 10.04 |
| | DARNet [5] | | 0.08 | 64.99 ± 13.67 | 64.15 ± 13.50 | 63.47 ± 12.34 |
| | DBPNet [6] | Dual | 0.88 | 64.77 ± 17.00 | 64.37 ± 16.28 | 60.92 ± 8.82 |
| | M-DBPNet [7] | | 1.32 / 1.00 / 0.88 | 68.90 ± 17.72 | 64.88 ± 15.33 | 61.03 ± 9.53 |
| | S²M-Former (ours) | | **0.06** | 70.64 ± 18.65 | 65.77 ± 15.58 | 65.49 ± 13.74 |

## 5 Results

We present a comprehensive evaluation of S²M-Former with five AAD methods across three datasets (KUL, DTU, AV-GC) in terms of architecture, size, and decoding accuracy under within-trial and cross-trial settings, as shown in Table 2 and Table 3. Our S²M-Former achieves the highest decoding accuracy in 11 out of 18 conditions, surpassing dual-branch counterparts DBPNet [6] (2/18, 11.1%), M-DBPNet [7] (2/18, 11.1%) and the single-branch SOTA DARNet [5] (3/18, 16.7%), while securing 83.33% Top-3 coverage (15/18), exceeding DBPNet (12/18, 72.2%) and M-DBPNet (11/18, 61.1%), and DARNet (9/18, 50%). Overall, the performance of dual-branch models tends to outperform single-branch models. Especially, S²M-Former excels in cross-trial performance, achieving SOTA results on the DTU dataset (e.g., 76.74% ± 9.96 for 2s) and AV-GC dataset (e.g., 70.64% ± 18.65 for 2s), demonstrating its robust generalization ability on unseen EEG data compared to recent methods. Notably, these results are achieved with only 0.06M parameters, using a fixed model size across all window lengths, unlike M-DBPNet, whose size varies by window (e.g., 1.32M for 2s, 1.00M for 1s and 0.88M for 0.1s). In comparison, S²M-Former is 14.7× smaller than DBPNet (0.88M) and 22× smaller than M-DBPNet (1.32M), yet consistently outperforms them. DBPNet and DARNet also have fixed sizes but with substantially more parameters than S²M-Former. Furthermore, decoding accuracy improves with larger decision windows, as extended context allows for more detailed attention estimation, consistent with prior research [14, 15].

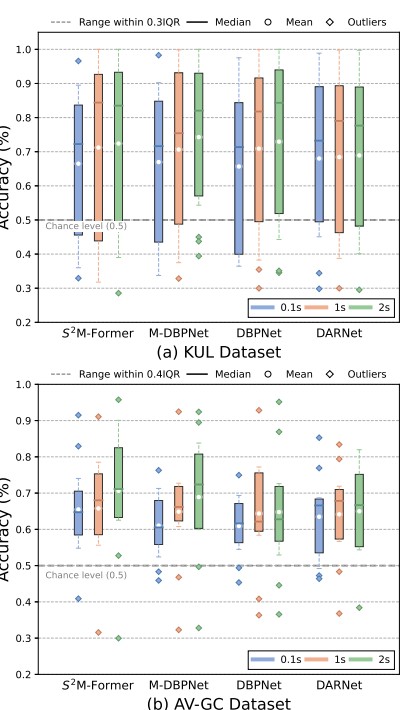

Figure 3: Comparison across all subjects on two datasets under cross-trial.

Although our model exhibits robust performance across all benchmarks under the cross-trial setting, as presented in Table 3, a notable performance gap remains when compared to the within-trial scenario. This disparity is especially pronounced on the KUL and AV-GC datasets, where elevated standard deviations reflect the increased difficulty of cross-trial generalization. Such challenges are largely attributed to greater intra-subject variability in EEG signals across trials and the inherent

Table 4: Ablation study and compare with recent SNN models on DTU dataset.

| Methods | Feature Embeddings | Param (M) | Time Steps | Within-trial | | Cross-trial | |
|---|---|---|---|---|---|---|---|
| | | | | 2s | 1s | 2s | 1s |
| S$^2$M-Former (proposed) | CSP+DE | 0.06 | 4 | **85.28 ± 6.01** | **82.87 ± 6.92** | **76.74 ± 9.96** | **75.75 ± 9.96** |
| SM-Former (ANN) | CSP+DE | 0.06 | — | 80.94 ± 5.66 | 80.30 ± 6.77 | 73.49 ± 8.09 | 73.48 ± 7.98 |
| CPLIF → LIF | CSP+DE | 0.06 | 4 | 84.13 ± 5.96 | 81.66 ± 6.66 | 75.67 ± 9.77 | 74.67 ± 9.82 |
| w/o SGCM & MPTM | CSP+DE | 0.05 | 4 | 82.98 ± 6.25 | 80.96 ± 7.23 | 74.81 ± 9.93 | 74.12 ± 9.52 |
| Spatial Branch | CSP | 0.04 | 4 | 82.28 ± 7.13 | 80.05 ± 6.69 | 73.58 ± 10.21 | 73.19 ± 9.71 |
| Frequency Branch | DE | 0.01 | 4 | 70.48 ± 7.78 | 69.07 ± 6.78 | 70.11 ± 11.57 | 68.98 ± 8.00 |
| QKFormer [39] | DE | 0.29 | 4 | 69.54 ± 8.80 | 68.84 ± 7.46 | 69.67 ± 9.84 | 67.83 ± 8.85 |
| Spike-driven Transformer [35] | DE | 0.37 | 4 | 69.44 ± 8.11 | 68.70 ± 7.30 | 68.29 ± 8.71 | 68.06 ± 8.74 |
| Spikformer [34] | DE | 0.37 | 4 | 67.28 ± 7.93 | 67.15 ± 6.52 | 69.27 ± 8.41 | 65.31 ± 9.03 |

challenge of generalizing to unseen data (zero-shot conditions [44]). To assess model robustness, we conducted a visualization analysis covering three decision windows and all subjects. While prior studies [7, 42] typically focused on a single decision window for their models, we extend the evaluation across multiple models and temporal ranges. As illustrated in Figure. 3, box plots show that subject-level performance is more dispersed in the cross-trial scenario, with several subjects falling below chance level (CL = 50%) [45] across all methods. Based on interquartile range (IQR) and the count of below-CL outliers, S$^2$M-Former exhibits more consistent behavior across datasets and decision windows, suggesting favorable robustness under zero-shot settings. More detailed statistics and analysis are provided in Appendix A.6.

We further conducted an ablation study to analyze our proposed model from two perspectives, as shown in Table 4. First, we replaced the spike-driven components in S$^2$M-Former with a pure ANN structure, termed SM-Former (see Appendix A.5 for detailed conversion steps). This modification led to a notable drop in decoding accuracy across all settings and time windows. Specifically, in the within-trial setting with the 2-second time window, accuracy dropped by 4.34%, from 85.28% (SD: 6.01) to 80.94% (SD: 5.96), highlighting the effectiveness of our SNN-friendly components in learning representations through spike-driven dynamics. From another perspective, we first validated the effect of the CPLIF neurons by replacing them with standard LIF neurons. This further degraded performance, with accuracy dropping from 76.25% (SD: 9.79) to 75.30% (SD: 9.77) for 2-second decision windows, and 75.75% (SD: 9.96) to 74.67% (SD: 9.82) for 1-second under the cross-trial setting. We then removed SGCM and MPTM entirely, as SGCM's output serves as the input for MPTM, and retained only the concatenation for fusion. The removal of this combination led to further degradation in decoding accuracy and an increase in standard deviation. This empirical outcome substantiates our initial hypothesis that the integration of complementary learning modules can enhance model performance. Finally, we compared the performance of the individual branches (Spatial- and Frequency-Branch) with the S$^2$M-Former. The individual branches performed both worse, with the spatial branch performing better than the frequency branch. Notably, under the same time step $T_S = 4$, the frequency branch surpasses three transformer-based SNN models, QKFormer [39] (0.29M), Spike-driven Transformer (SDT) [35] (0.37M), and Spikformer [34] (0.37M). See Appendix A.6 for more results. The above impression can be attributed to two main factors: (1) the spike-driven hierarchical architecture within each branch helps capture both long-range dependencies and fine-grained details efficiently; (2) the symmetric structure encourages complementary learning between the spatial and frequency domains, enabled by biologically plausible token-channel mixers.

Table 5 analyzes the efficiency advantages with detailed theoretical energy consumption provided in Appendix A.7. Under 2-second windows, S$^2$M-Former achieves exceptional efficiency in dual-branch setups: 93% smaller size compared to DBPNet (0.06M vs. 0.88M, 14.7× reduction), 82.8% lower energy than DBPNet (0.0779 mJ vs. 0.4526 mJ, 5.8× re-

Table 5: Estimation of sizes and operational counts across *single-branch* and dual-branch models under cross-trial setting.

| Model | SNN | Params (M) ↓ | FLOPs (G) ↓ | SOPs (G) ↓ | Energy (mJ) ↓ |
|---|---|---|---|---|---|
| *DARNet* | ✗ | 0.08 | 0.0054 | — | 0.0247 |
| *QKFormer* | ✓ | 0.29 | 0.0015 | 0.0160 | 0.0212 |
| *SDT* | ✓ | 0.37 | 0.0015 | 0.0227 | 0.0272 |
| *Spikformer* | ✓ | 0.37 | 0.0015 | 0.0065 | 0.0126 |
| DBPNet | ✗ | 0.88 | 0.0984 | — | 0.4526 |
| M-DBPNet | ✗ | 1.32 | 0.1068 | — | 0.4913 |
| SM-Former | ✗ | 0.06 | 0.0243 | — | 0.1116 |
| S$^2$M-Former | ✓ | 0.06 | 0.0112 | 0.0293 | 0.0779 |

duction), benefiting from its efficient lightweight architecture and spike-driven mechanism. Compared to its ANN counterpart, 53.9% fewer FLOPs than SM-Former (0.0112 G vs. 0.0243 G). Although

S$^2$M-Former's energy consumption (0.0779 mJ) is higher than that of single-branch SNNs, this is due to the additional FLOPs from the dual-branch input embeddings. When considering synaptic operations (SOPs), our model generates comparable SOPs to other single-branch SNNs (0.0293 G vs. 0.0227 G, 0.0160 G, and 0.0065 G), indicating the effectiveness of our module design. Although DARNet shows energy efficiency, it still substantially lags behind S$^2$M-Former in performance level.

We further conduct cross-subject (LOSO) validation on the KUL and DTU datasets to evaluate the model's robustness across subjects, consistent with previous methods [5, 43]. This setting presents a more challenging and realistic evaluation scenario, as the model must generalize to entirely unseen subjects. As

Table 6: Leave-one-subject-out cross-validation experiments.

| Model | KUL | | DTU | |
|---|---|---|---|---|
| | 2-second | 1-second | 2-second | 1-second |
| DARNet | 74.65 ± 15.77 | 73.72 ± 15.09 | 59.12 ± 4.43 | **58.48 ± 4.67** |
| DBPNet | 74.82 ± 13.40 | 71.77 ± 14.07 | 56.83 ± 5.00 | 54.76 ± 4.38 |
| M-DBPNet | 72.83 ± 12.49 | 71.72 ± 13.35 | 54.48 ± 5.57 | 53.56 ± 6.16 |
| S$^2$M-Former | **75.75 ± 13.43** | **74.37 ± 12.57** | **59.75 ± 5.25** | 57.70 ± 4.21 |

summarized in Table 6, our proposed S$^2$M-Former consistently outperforms all competing methods across both 1-second and 2-second decision windows. Notably, it achieves the highest accuracy of 75.75% on KUL and 59.75% on DTU under the 2-second window, while maintaining competitive performance even in the more context-constrained 1-second window. These results underscore the strong generalization ability of S$^2$M-Former across broad experimental settings, validating its effectiveness in real-world AAD scenarios where inter-subject differences are prominent.

## 6  Conclusion

In this work, we introduce S$^2$M-Former, an efficient spiking symmetric mixing network designed to address key challenges in AAD tasks. By integrating spike-driven hierarchical modeling with spatial-frequency complementary learning, our model achieves high accuracy while maintaining computational efficiency. S$^2$M-Former not only delivers competitive state-of-the-art performance, but also demonstrates robust generalization across diverse evaluation settings, including unseen data and subjects. Furthermore, it outperforms recent dual-branch models with fewer parameters and lower energy consumption, while avoiding isolated learning paradigms. These advantages make S$^2$M-Former a promising neuromorphic solution for energy-efficient auditory attention decoding.

## Ethics Statement

In this work, we do not generate new EEG data, nor do we perform experiments on human subjects. We use the three publicly available KUL, DTU, and AV-GC-AAD datasets without any restrictions. The public download links are as follows:

- KUL[19, 46]: `https://zenodo.org/records/4004271`
- DTU [20, 47]: `https://zenodo.org/records/1199011`
- AV-GC-AAD [21, 41, 48]: `https://zenodo.org/records/11058711`

## Acknowledgments

This work is supported by the National Science and Technology Innovation 2030 Major Project (No. 2025ZD0215501), the National Natural Science Foundation of China (No. 82272114) and the Shenzhen Science and Technology Program (No. ZDSYS20230626091203008).

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

# A    Technical Appendices

## A.1    EEG Feature Embedding

We segment the evoked EEG into decision windows employing the sliding window method. Each decision window $E$ is represented as an $C \times T$ matrix, which is formed as $E = [e_1, e_2, ..., e_t] \in \mathbb{R}^{C \times T}$, where $T$ denotes the length of the window, and $C$ represents the number of EEG channels. Each column $e_i$ corresponds to the EEG signal across all channels at a specific time sample $t$.

As shown in Figure. 2, our proposed S$^2$M-Former adopts a dual-branch architecture to process two types of feature embeddings derived from evoked EEG data $E$. For clarity, we refer to them as the *spatial branch* and the *frequency branch*, and elaborate on the corresponding input embeddings $E_S$ and $E_F$ in the following sections.

For the *spatial branch*, the CSP algorithm is adopted [12], which has demonstrated the effectiveness of extracting primary spatial features for AAD tasks. Specifically, CSP finds the optimal spatial filters using covariance matrices [13]. The processing can be formed as $E_S = CSP(E) \in \mathbb{R}^{d_n \times T}$, where $d_n$ is the number of components to decompose EEG signals. Similar to the DARNet [5], we select the $d_n = C = 64$. It is worth noting that some previous works [5, 6, 7], have regarded the CSP-extracted branch as the temporal branch. Although $E_S$ retains the sampling points $T$, we prefer to define it as the spatial-(temporal) branch.

For the *frequency branch*, we firstly decompose the EEG signals into five frequency bands (from $\delta$ band to $\gamma$ band), considering the power spectrum of different frequencies, we further extract the differential entropy (DE) feature from each frequency band and then project them onto a size of $H \times W$ 2D topological maps to utilize their topological patterns [14, 15]. Specifically, in the implementation, $H = W = 32$. Taking a specific frequency band as an example, the spectral feature can be formed as $E_{Fi} = DE(E_i) \in \mathbb{R}^{1 \times H \times W}$, where $i$ denotes the $i$th frequency band. Finally, we concatenate five maps into a $E_F = [E_{F1}, E_{F2}, ..., E_{F5}] \in \mathbb{R}^{5 \times H \times W}$.

## A.2    Spiking Neuron

To achieve *intra-module communication*, we employ the Leaky Integrate-and-Fire (LIF) as the spiking neuron. The following discrete-time equations govern the dynamics of an LIF neuron:

$$H[t] = V[t-1] - \frac{1}{\tau}\left((V[t-1] - V_{reset})\right) + X[t], \tag{20}$$

$$S[t] = \Theta\left(H[t] - V_{th}\right), \tag{21}$$

$$V[t] = H[t]\left(1 - S[t]\right) + V_{reset}S[t], \tag{22}$$

where $\tau$ is the membrane time constant, $X[t]$ is the input current at time step $t$, $V_{reset}$ is the reset potential, and $V_{th}$ is the firing threshold. Eq. (20) describes the membrane potential update, while Eq. (21) models spike generation, where $\Theta(v)$ is the Heaviside step function: if $H[t] \geq Vth$, $\Theta(v) = 1$, indicating a spike; otherwise, $\Theta(v) = 0$. $S[t]$ represents whether the neuron fires a spike at time step $t$. Eq. (22) defines the reset of the membrane potential, where $H[t]$ and $V[t]$ denote the membrane potential before and after spike generation at time step $t$, respectively.

To enhance membrane potential awareness across *inter-module connections* within the S$^2$M-Former, we propose the CPLIF neuron. The membrane potential update rule for CPLIF can refer to Eq. (1), which extends the standard PLIF by assigning a learnable membrane time constant to each individual channel. This design enables finer control over temporal dynamics at each discrete time, allowing more expressive and adaptive modeling. Here, $\tau_l[c]$ is the softmax activation function to perceive channel interaction, and also ensures that $\frac{1}{\tau_l[c]} \in (1, \infty)$. The firing and reset equations of the CPLIF neuron are the same as the Eq. (21) and Eq. (22) of the standard LIF neuron.

## A.3    Datasets

1) KUL [19, 46]: The dataset consists of 64-channel EEG recordings from 16 normal-hearing subjects, collected using a BioSemi ActiveTwo system at a sampling rate of 8192 Hz. Each subject was instructed to focus on one of two simultaneous speakers. The auditory stimuli, consisting of four Dutch short stories narrated by three male Flemish speakers, were presented under two conditions:

Table 7: Implementation details for KUL, DTU, and AV-GC datasets.

| Datasets | KUL | DTU | AV-GC |
|---|---|---|---|
| Training Epochs | | 300 | |
| Batch Size | 32 (Subject-dependent setting) / 128 (Subject-independent setting) | | |
| Optimizer | | Adam | |
| Learning Rate (Within-trial) | 1e-3 | 5e-4 | 5e-4 |
| Learning Rate (Cross-trial) | | 2e-4 | |
| Learning Rate (Cross-subject) | | 2e-3 | |
| Weight Decay | | 1e-2 | |
| Spiking Neuron | LIF ($\tau$=2.0, $V_{threshold}$=1.0), CPLIF ($\tau$=2.0, $V_{threshold}$=1.0) | | |
| Time Steps | | 4 | |
| Surrogate Function | | Atan ($\alpha$=5.0) | |
| LR Scheduler | | Cosine Annealing WarmRestarts | |
| Seed | | 200 | |
| GPUs | | RTX 4090 | |

dichotic (dry) presentation, with one speaker per ear, and head-related transfer function (HRTF) filtered presentation simulating speech from 90° to the left or right. The stimuli were delivered through in-ear headphones, filtered at 4 kHz, and presented at 60 dB. Each subject participated in 8 trials, each lasting 6 minutes, for a total of 12.8 hours of EEG data.

2) DTU [20, 47]: The dataset contains 64-channel EEG recordings from 18 normal-hearing subjects, collected using a BioSemi ActiveTwo system at a sampling rate of 512 Hz. Each subject was instructed to focus on one of two simultaneous speakers, who narrated Danish audiobooks through ER-2 earphones set at 60 dB. The audiobooks, narrated by three male and three female speakers, were presented at a 60° angle relative to the subject's frontal position. The auditory stimuli were presented in a mixed speech format with varying reverberation levels. Each subject completed 60 trials, each lasting 50 seconds, resulting in a total of 15 hours of EEG data.

3) AV-GC-AAD [21, 41, 48]: The audiovisual, gaze-controlled auditory attention dataset consists of EEG recordings from 16 normal-hearing subjects who focused on one of two competing talkers located at ± 90° relative to the subject. EEG data were recorded using a 64-channel BioSemi ActiveTwo system. The auditory stimuli, consisting of Dutch science podcasts, were presented through insert earphones using HRTF to simulate spatial separation. The experiment involved 4 conditions—no visuals, static video, moving video, and moving target noise—each with 2 trials lasting 10 minutes. Detailed information on each condition is available in Rotaru et al [21]. The visual setups varied to explore the effect of gaze on auditory attention, with the to-be-attended speaker switching sides after 5 minutes to simulate a spatial attention shift. EEG data from subjects #2, #5, and #6 were excluded due to a lack of consent for public sharing. Subjects #1 and #3 were further excluded due to missing condition 4 recordings, leaving 11 subjects for analysis. For the cross-trial setting, a controlled paradigm with four conditions is used: conditions 1 (auditory-only) and 2-3 (audio-visually static/moving videos) for training, and condition 4 (incongruent moving-target noise) for testing. This setting evaluates model robustness against cross-modal conflicts by examining domain shifts between audiovisual-related (training) and mismatched (testing) conditions, and analyzing how audiovisual congruency impacts generalization in AAD models.

## A.4 Experimental Setup

We use the Adam optimizer with a learning rate to minimize the cross-entropy loss, setting the batch size to 32 (or 128 for subject-independent setting) and training for 300 epochs, more details are shown in the Table 7. The hidden dimension $D$ in S$^2$M-Former is set to 8. An early stopping criterion is applied if no significant improvement in the loss function is observed over 25 consecutive epochs, automatically halting training. The best model is saved based on both validation loss and accuracy, and the model with the best performance is loaded for final evaluation. To ensure fairness, our proposed model and all reproducible baselines are trained following their originally reported optimization strategies, including the specific choice of optimizer, learning rate, and related hyperparameters. In addition, the same random seed is applied across all experiments to ensure reproducibility and

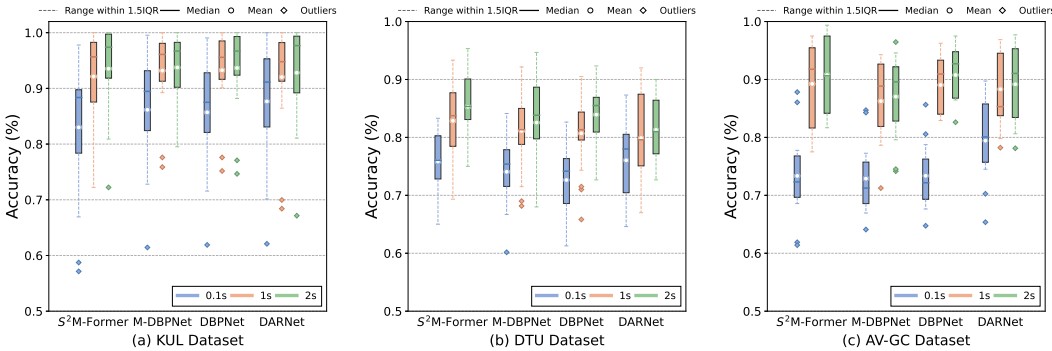

Figure 4: Visualization comparison across all subjects on three datasets under within-trial settings.

fairness. All models are constructed and implemented using the PyTorch and SpikingJelly [30] frameworks, and all experiments are conducted on an NVIDIA GeForce RTX 4090 GPU.

### A.5 Conversion Steps from S$^2$M-Former to SM-Former

To enable a fair and controlled comparison between spiking and non-spiking models, we construct a matched ANN counterpart to our proposed **S$^2$M-Former**, termed **SM-Former**. SM-Former retains the overall architectural structure, layer depth, and parameter budget of S$^2$M-Former, while systematically replacing all spike-driven components with their standard ANN analogues. The conversion steps are detailed below:

- **CPLIF Neuron Removal:** All channel-wise parametric PLIF (CPLIF) neurons are removed, eliminating inter-module temporal spiking dynamics.
- **Branch Encoder Conversion:** In both the spatial (SBE) and frequency (FBE) branches, all spiking neuron layers (e.g., LIF-Conv-BN) are replaced by ANN layers (e.g., Conv-BN-ReLU).
- **SCSA Replacement:** The Spiking Channel Self-Attention (SCSA) module is replaced with a standard ANN-based channel-wise attention, employing QKV projections and softmax operations.
- **SMSC Conversion:** All spiking preprocessing blocks in the Spiking Multi-Scale Convolution (SMSC) module are replaced with conventional convolutional layers, such as DWConv-BN-ReLU.
- **SGCM Conversion:** In the Spiking Gated Channel Mixer (SGCM), blocks like LIF-Conv-BN and LIF-DWConv-BN-LIF are replaced with Conv-BN-ReLU and DWConv-BN-ReLU, respectively.
- **MPTM Simplification:** The Membrane Potential-aware Token Mixer (MPTM) retains its structure but eliminates all spiking operations, replacing spike-gating and temporal fusion mechanisms with standard ANN counterparts.

This conversion preserves the representational capacity and computational structure, ensuring that the performance differences between S$^2$M-Former and SM-Former arise from the use of spike-driven versus non-spiking operations, rather than from architectural or parameter count differences.

### A.6 Comprehensive Results

Figure. 5 presents a comparative statistics of model performances under within-trial and cross-trial settings. An overarching trend emerges from both settings: dual-branch architectures (e.g., DBPNet, M-DBPNet, and S$^2$M-Former) consistently outperform single-branch models (e.g., MBSSFCC and DARNet), highlighting the strength of leveraging multiple feature representations in AAD tasks. In the within-trial scenario (Figure. 5a), S$^2$M-Former achieves the highest decoding accuracy in 4 out of 9 conditions, outperforming DBPNet [6] and DARNet [5], which each lead in only 2 conditions, indicating its superior performance on subject-seen data across all datasets. Moreover, in the more challenging cross-trial setting (Figure. 5b), S$^2$M-Former exhibits clear superiority, attaining top-1 accuracy in 7 out of 9 conditions, 3.5 times as many as all competing models combined, highlighting its strong generalization across subjects.

Table 8: Ablation study and compare with recent SNN models on KUL dataset.

| Methods | Feature Embeddings | Param (M) | Time Steps | Within-trial | | Cross-trial | |
| --- | --- | --- | --- | --- | --- | --- | --- |
| | | | | 2s | 1s | 2s | 1s |
| $S^2$M-Former (proposed) | CSP+DE | 0.06 | 4 | **93.71 ± 8.14** | **92.27 ± 8.66** | **72.39 ± 25.21** | **71.22 ± 25.97** |
| SM-Former (ANN) | CSP+DE | 0.06 | — | 91.38 ± 9.71 | 90.65 ± 10.80 | 69.46 ± 23.75 | 66.69 ± 23.52 |
| CPLIF → LIF | CSP+DE | 0.06 | 4 | 92.77 ± 9.01 | 91.60 ± 9.47 | 71.23 ± 24.37 | 69.50 ± 25.24 |
| w/o SGCM & MPTM | CSP+DE | 0.05 | 4 | 91.99 ± 10.27 | 90.67 ± 10.36 | 70.55 ± 25.88 | 69.04 ± 25.89 |
| Spatial Branch | CSP | 0.04 | 4 | 90.91 ± 9.22 | 89.68 ± 10.01 | 69.87 ± 24.20 | 67.79 ± 25.06 |
| Frequency Branch | DE | 0.01 | 4 | 89.15 ± 10.54 | 88.90 ± 10.03 | 70.18 ± 22.09 | 68.09 ± 21.42 |
| QKFormer [39] | DE | 0.29 | 4 | 85.42 ± 11.36 | 84.87 ± 12.82 | 65.96 ± 20.97 | 65.24 ± 23.36 |
| Spike-driven Transformer [35] | DE | 0.37 | 4 | 86.24 ± 11.20 | 84.81 ± 13.51 | 64.94 ± 22.78 | 64.83 ± 21.17 |
| Spikformer [34] | DE | 0.37 | 4 | 83.16 ± 12.36 | 82.90 ± 13.46 | 64.11 ± 22.00 | 63.06 ± 22.79 |

Table 9: Ablation study and compare with recent SNN models on AV-GC dataset.

| Methods | Feature Embeddings | Param (M) | Time Steps | Within-trial | | Cross-trial | |
| --- | --- | --- | --- | --- | --- | --- | --- |
| | | | | 2s | 1s | 2s | 1s |
| $S^2$M-Former (proposed) | CSP+DE | 0.06 | 6/4 | **91.83 ± 6.66** | **89.24 ± 7.59** | **70.64 ± 18.65** | 65.77 ± 15.58 |
| SM-Former (ANN) | CSP+DE | 0.06 | — | 90.66 ± 6.84 | 88.07 ± 7.10 | 68.83 ± 17.50 | **66.52 ± 13.75** |
| CPLIF → LIF | CSP+DE | 0.06 | 6/4 | 91.23 ± 6.78 | 88.68 ± 7.57 | 69.54 ± 18.38 | 65.19 ± 16.03 |
| w/o SGCM & MPTM | CSP+DE | 0.05 | 6/4 | 90.72 ± 6.63 | 88.03 ± 7.45 | 68.84 ± 16.24 | 64.51 ± 15.90 |
| Spatial Branch | CSP | 0.04 | 6/4 | 89.12 ± 7.31 | 87.33 ± 7.41 | 68.02 ± 15.00 | 63.85 ± 14.55 |
| Frequency Branch | DE | 0.01 | 6/4 | 80.42 ± 8.15 | 78.21 ± 8.24 | 63.26 ± 15.08 | 62.25 ± 15.41 |
| QKFormer [39] | DE | 0.29 | 6/4 | 77.87 ± 6.42 | 76.90 ± 8.81 | 63.54 ± 13.39 | 60.20 ± 13.02 |
| Spike-driven Transformer [35] | DE | 0.37 | 6/4 | 80.15 ± 7.60 | 78.49 ± 7.85 | 62.53 ± 15.34 | 59.48 ± 11.98 |
| Spikformer [34] | DE | 0.37 | 6/4 | 75.05 ± 7.66 | 73.09 ± 6.91 | 64.16 ± 14.69 | 63.32 ± 14.23 |

Remarkably, $S^2$M-Former achieves this with only 0.06M parameters, significantly outperforming larger models, such as MBSSFCC [15] (16.81M) and DBPNet (0.88M), thereby demonstrating exceptional parameter efficiency. These results collectively reinforce the effectiveness of $S^2$M-Former in delivering robust and efficient AAD solutions. Moreover, while DBPNet's top-1 count drops from 2 (within-trial) to 0 (cross-trial), M-DBPNet [7] maintains stable performance and outperforms DBPNet under cross-trial evaluation, corroborating prior findings on its stronger generalization ability.

We further provide the visualization analysis across all subjects on three datasets under within-trial settings, as shown in Figure. 4. In terms of this setting, we found that the distributions of decoding accuracies across all models and datasets are relatively concentrated, so we adopt the conventional 1.5 IQR rule which can effectively distinguish true variability from extreme outliers without excessively compressing the distribution range, thereby preserving the subtle differences among methods while avoiding misleading impressions of overstability. Consistent with findings under the cross-trial setting, our $S^2$M-Former exhibits the fewest outliers,

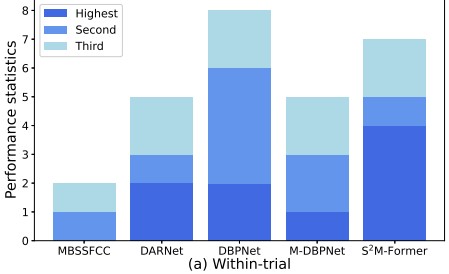

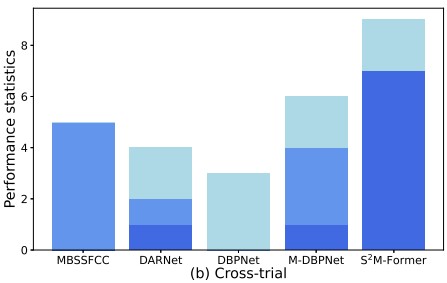

Figure 5: Performance statistics across all models under within- and cross-trial.

particularly demonstrating its advantage under the 1-second and 2-second decision windows and DTU dataset. We further supplement the results with a comparison across all subjects on the DTU dataset under the cross-trial setting, as shown in Figure. 6. For the vast majority of methods, the decoding accuracy of all subjects exceeds the chance level. Notably, consistent with the observations in Figure. 3, our method exhibits fewer outliers compared to other baselines, demonstrating superior robustness and generalization capability. A phenomenon worth discussing is that the performance degradation from within-trial to cross-trial on the DTU dataset is less severe compared to that observed on the KUL and AV-GC datasets. We attribute this to the larger number of trials per subject in the DTU dataset, which significantly enhances training stability and improves testing robustness under the zero-shot condition. In contrast, datasets with fewer trials per subject are more prone to poor generalization, potentially resulting in below-chance performance on the test set.

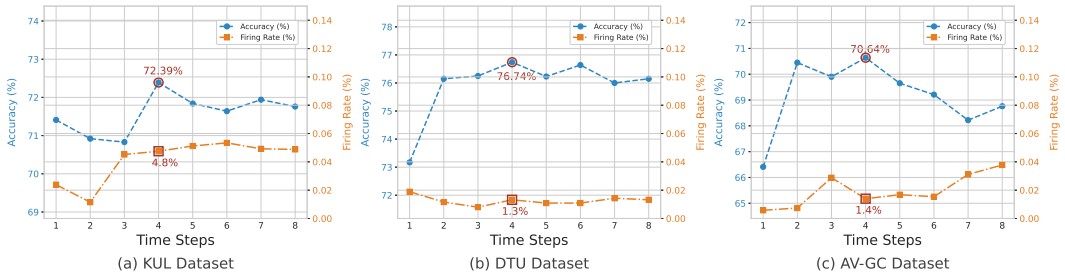

Figure 8: Visualization of the accuracy and firing rate change with the number of time steps $T_S$, where the blue line is the accuracy and the red line is the firing rate.

We conducted comprehensive ablation studies on the KUL dataset, as shown in Table 8. The proposed $S^2M$-Former consistently outperforms its ANN counterpart (SM-Former) across all settings, demonstrating the advantage of our spike-driven design. For example, under the 2-second within-trial condition, $S^2M$-Former achieves 93.71% (SD: 8.14), surpassing SM-Former's 91.38% (SD: 9.71). Replacing the CPLIF neurons with standard LIF leads to consistent performance drops, validating the benefit of channel-wise adaptive time constants (e.g., 71.22% $\rightarrow$ 69.50% under 1s cross-trial). Removing SGCM and MPTM further degrades results, highlighting their complementary roles in biologically plausible token-wise and channel-wise modeling. Branch-level ablations show that the Spatial branch performs better in within-trial settings (e.g., 90.91% vs. 89.15%), while the Frequency branch is more effective in cross-trial scenarios (e.g., 70.18% vs.

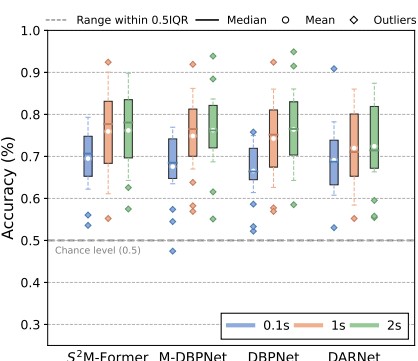

Figure 6: Comparison across all subjects on the DTU dataset under cross-trial.

69.87%). Notably, despite having only 0.01M parameters, the Frequency-branch alone matches or exceeds recent SNN models such as QKFormer [39], Spike-driven Transformer [35], and Spikformer [34] (e.g., 89.15% vs. 85.52%, 86.24% and 83.16%), with more than 29 × fewer parameters (0.01 vs. 0.29M). Ablation results on the AV-GC dataset (Table 9) show generally consistent trends. Two exceptions are observed: (1) SM-Former slightly outperforms $S^2M$-Former under the 1-second cross-trial condition; and (2) the Spatial-branch consistently outperforms the Frequency-branch across all settings, aligning with observations from the DTU dataset. The above analyses further support the robustness of $S^2M$-Former and the generalizability of its key components across datasets.

We evaluated the effect of time steps on model performance under the cross-trial setup with a 2-second decision window. Figure. 8 shows the classification accuracy and corresponding average firing rate across all subjects for three datasets, with time steps extending from 1 to 8. Specifically, $S^2M$-Former achieves the highest accuracy of 72.39% with a firing rate of 4.8% on the KUL dataset, 76.25% with 1.3% on DTU, and 70.64% with 1.4% on AV-GC. With the initial increase in time steps, the accuracy generally improves (e.g., from 3 to 4 time steps on KUL, 1 to 4 time steps on DTU, and 1 to 4 time steps on AV-GC). However, beyond 4 time steps, the performance gain becomes marginal, while the cumulative firing rate continues to rise, which is particularly evident on the KUL and AV-GC datasets. This observation suggests that further increasing the temporal resolution yields limited improvements in accuracy but incurs additional computational cost. Therefore, we choose 4 time steps as a trade-off, striking a favorable balance between performance and energy efficiency.

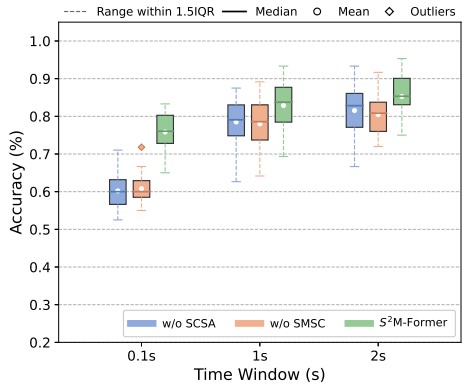

Figure 7: Ablation studies on SCSA and SMSC modules on DTU dataset under within-trial.

Figure. 7 presents the ablation results of the hierarchical representation learning modules in S$^2$M block, consisting of SCSA for modeling long-range global dependencies and SMSC for capturing local fine-grained patterns. Even when either module is removed, the subsequent SGCM and MPTM layers are retained to ensure meaningful cross-branch complementary fusion learning. Under the shortest time window (0.1s), SCSA (60.79 ± 4.23) outperforms SMSC (60.19 ± 4.60). However, the combination of SCSA and SMSC significantly boosts accuracy to 75.84 ± 5.46, yielding a 15.05% improvement over SCSA alone, highlighting the comprehensive strengths of global and local representations. At the 1s window, SMSC slightly outperforms SCSA (78.51 ± 6.64 vs. 78.01 ± 6.93), suggesting that as temporal context extends, modeling local dynamics becomes increasingly important. Their integration further boosts performance to 82.87 ± 6.92. At the 2s window, SMSC shows advantage over SCSA (81.54 ± 6.66 vs. 80.44 ± 5.47). Combining both modules achieves the highest accuracy of 85.28 ± 5.61, improving by 3.74% over SMSC and 4.84% over SCSA, validating the effectiveness of our S$^2$M-Former architecture.

To further assess the energy efficiency and dynamic characteristics of S$^2$M-Former, we analyze the average spike firing rates within its spatial branch, frequency branch, and fusion module (SGCM & MPTM) under cross-trial settings on three datasets (Figure. 9). All components exhibit low firing rates (<0.08), confirming the model operates with the sparse nature of spiking activations, which minimizes computation and improves overall efficiency. Among the branches, the spatial branch generally exhibits the highest firing rates, consistent with ablation results (Table 4, 8, and 9), where spatial cues contribute more substantially to performance in most conditions. Interestingly, on the KUL dataset with a 2s window, the frequency branch surpasses

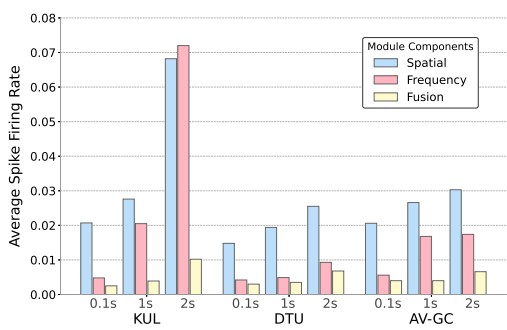

Figure 9: Firing rate analysis of S$^2$M-Former across all datasets.

the spatial branch in firing activity. This corresponds with its performance under this setting and suggests that S$^2$M-Former can adapt the firing behavior of its branches in response to the informativeness of spatial or frequency cues in the input dynamically. As decision windows extend, firing rates rise due to increased input information, yet sparsity is still preserved. Notably, the fusion module consistently shows the lowest firing rates yet remains crucial for performance (per ablation studies). This phenomenon suggests that the fusion module can effectively coordinate and integrate abstracted representations from branches with minimal spiking activity. Such a mechanism might mimic the sparsely active characteristics observed in integration layers [49] of biological neural systems.

### A.7 Theoretical Calculation of Energy Consumption

**For ANNs**, the theoretical energy consumption is calculated by multiplying the total number of multiply-accumulate (MAC) operations by the energy per MAC operation on specified hardware. Using the fvcore library [50] to compute floating-point operations (FLOPs), the energy consumption can be expressed as:

$$E_{ANN} = E_{MAC} \times \sum_{l=1}^{L} FLOP^l \tag{23}$$

where FLOP denotes the number of MAC operations in layer $l$, and $E_{MAC} = 4.6pJ$ represents the energy cost per MAC operation on 45nm hardware [51].

**For SNNs**, energy consumption involves both MAC operations and spike-driven accumulate (AC) operations [31, 52]. The number of synaptic operations (SOPs) is calculated as:

$$SOP^l = fr^{l-1} \times FLOP^l, \tag{24}$$

where $fr^{l-1}$ is the firing rate of spiking neuron layer $l-1$, which can be computed as:

$$fr^{l-1} = \frac{1}{T \times N} \sum_{t=1}^{T} \sum_{i=1}^{N} s_i^{l-1}(t), \tag{25}$$

where $T$ is the total number of time steps, $N$ is the number of neurons in layer $l-1$, and $s_i^{l-1}(t)$ denotes the spike output (0 or 1) of the $i$-th neuron at time step $t$. $FLOP^l$ refers to the equivalent MAC operations of layer $l$, and $SOP^l$ is the number of spike-based AC operations (SOPs). Assuming the MAC and AC operations are performed on the 45nm hardware [51], the consumption of the spiking transformer can be calculated as follows:

$$E = E_{MAC} \times \left(FLOP_{Conv}^1\right) + E_{AC} \times \left(\sum_{i=2}^{N} SOP_{Conv}^i + \sum_{j=1}^{M} SOP_{SSA}^j\right), \qquad (26)$$

where $SOP_{Conv}$ represents the SOPs of a convolution or linear layer, and $SOP_{SSA}$ represents the SOPs of an SSA module, $FLOP_{Conv}^1$ represents the FLOPs of the first layer before encoding input frames into spikes. $N$ is the total number of convolution layers and linear layers, and $M$ is the number of SSA modules. During model inference, several cascaded linear operation layers such as convolution, linear, and BN layers, can be folded into one single linear operation layer [39], still enjoying the AC-type operations with a spike-form input tensor.

The energy consumption calculation for our S$^2$M-Former also follows Eq.(26). Specifically, the MAC operations are primarily generated by the first convolution layer of the branch-specific spiking encoders SBE and FBE, while the remaining parts involve SOP calculations.

## A.8 Future Work and Limitation

Our proposed S$^2$M-Former is a lightweight and low-power spiking neural network to tackle AAD tasks, which has demonstrated its effectiveness through comprehensive experiments. Notably, it utilizes only 0.06M parameters, significantly outperforming recent network models in terms of parameter efficiency. In future research, we aim to further unlock the full potential and scalability of S$^2$M-Former. Additionally, SNNs are known for their spatial-temporal dynamics. In our solution, S$^2$M-Former achieves SOTA feature-oriented representation inter-mixing, but the utilization of CSP and DE features inevitably disrupts the inherent temporal dynamics. In future work, we are committed to exploring how to model the temporal dynamics inherent in EEG data.

Moreover, SNNs are particularly appealing for their compatibility with neuromorphic hardware due to their sparse and event-driven nature. In line with recent influential SNN studies in other realms (e.g., CV, NLP and Speech), we have reported the theoretical energy consumption of S$^2$M-Former and compared it against other SNN-based approaches. This analysis further validates the energy efficiency of our model. To bridge algorithm and hardware co-design, we are currently conducting hardware simulations of S$^2$M-Former on brain-inspired neuromorphic platforms. This line of work will help validate the real-world deployment potential of our model on dedicated low-power chips.

