# OpenReview forum: "S$^2$M-Former: Spiking Symmetric Mixing Branchformer for Brain Auditory Attention Detection"
_NeurIPS.cc/2025/Conference — NeurIPS 2025 poster_

### Official Review · Reviewer_aRKz · 2025-06-24

**Clarity:** 3
**Significance:** 3
**Originality:** 3
**Rating:** 5
**Confidence:** 3

**Summary:**

Authors proposes $S^2M$-Former, a spiking neural network (SNN) model for auditory attention detection based on EEG signals. This method is the first spiking framework that is capable of mixing features extracted from spatial-temporal EEG dynamics and brain topographical maps. Compare to the common yet naive concatenation of features from both branches, the authors take the path of hierarchical integration. By leveraging spiking neurons, $S^2M$-Former reduces the parameter count and energy consumption.

**Questions:**

1. What is the difference between $S^2M$-Former and SM-Former? Is it a simple deletion of spiking neurons? More details about SM-former will be helpful.
2. Which part of the proposed framework contribute most to capture the long-range dependencies?
3. What might be the reason behind the narrowed advantage over prior non-SNN methods (e.g., DBPNet) in the KUL benchmark?

**Ethical Concerns:**

["NO or VERY MINOR ethics concerns only"]

**Final Justification:**

This paper mainly proposes a hierarchical integration of EEG and topographical maps for more robust auditory attention detection. The reported results on several benchmarks show notable improvements over prior works. During the rebuttal phase, the authors answer and clarify the questions raised by reviewers and make the paper much stronger by including the implementation details, computational efficiency, and analysis of the experimental results. Given the merit of the proposed method, I maintain a positive rating.

**Limitations:**

yes

**Quality:**

3

**Strengths And Weaknesses:**

### Strengths:
1. Well motivated and carefully designed components. The functionality of each block is clearly presented in the figure and their individual effectiveness is thoroughly discussed in the ablation study.
2. The comparison and benchmark clearly shows the strength of the proposed model.
3. Compared to prior works like Spikformer (SNN) and DBPNet (ANN), its improvement in efficiency is quite noticeable.
### Weaknesses:
1. Performance in terms of accuracy improvements is not consistent across different benchmarks.

---

> ### Author Rebuttal · Authors · 2025-07-28
>
> Dear Reviewer aRKz,
>
> Thanks for your constructive comments. We will address your concerns point by point in the following.
>
> ### **[Weakness]:**
>
> **1. Performance in terms of accuracy improvements is not consistent across different benchmarks.**
>
> Thank you for your valuable feedback. We acknowledge the performance variations observed across datasets and appreciate the opportunity to clarify.
>
> First, such variations may reflect **intrinsic differences across datasets**. As summarized in **Table 1**, the **KUL**dataset, **with fewer trials and longer recordings (8 trials × 360s)**, presents a relatively simpler decoding scenario and is more susceptible to **performance saturation**. In contrast, **DTU** (60 trials per subject with only 50s recordings) and **AV-GC-AAD** (audio-visual distraction with multi-condition) pose greater challenges due to increased inter-subject variability and realistic multisensory interference, thereby offering a more rigorous benchmark for model robustness.
>
> Second, our goal is not to optimize for a single dataset, but to demonstrate generalizability, consistently strong performance, and efficiency across **diverse AAD conditions**. As discussed in Section *Results* (Page 7, Lines 257–271) and Appendix A.5 (Fig. 5 and its discussion; Page 15, Lines 588–613),  S²M-Former achieves the highest accuracy in **11 out of 18** test conditions and achieves top-3 performance in 83.33%, consistently outperforming dual-branch baselines (e.g., DBPNet, M-DBPNet) in terms of **cross-dataset reliability and robustness**.
>
> While S²M-Former does not always achieve the highest accuracy on KUL within-trial decoding (Table 2), its strength becomes more evident in **cross-trial setups**, where it reaches the best accuracy under 1s windows and competitive results under 0.1s (Table 3). More importantly, under the most rigorous **leave-one-subject-out (LOSO)** evaluation (Table 6), S²M-Former achieves the best performance, highlighting its strong generalization across subjects.
>
> Importantly, S²M-Former achieves this with only 0.06M parameters, offering a 14.7× parameter reduction and 5.8× lower theoretical energy cost compared to DBPNet  (Table 5), while maintaining or exceeding its performance. Even in benchmarks where accuracy margins are narrower (e.g., KUL), S²M-Former delivers **comparable accuracy with significantly better efficiency**, which is particularly valuable for **real-world, low-power applications**.
>
> We will clarify this point further in the final version of the paper.
>
> ### **[Questions]:**
>
> **1. What is the difference between S²M-Former and SM-Former? Is it a simple deletion of spiking neurons?**
>
> Thank you for your question. To clarify, **SM-Former is NOT merely a simple deletion of spiking neurons replacement**, but rather a **rational** ANN-based version of S²M-Former, where all spike-driven components are systematically replaced with their standard ANN counterparts, while preserving the overall architectural structure and parameter count.
>
> Specifically:
>
> - We first remove all the proposed channel-wise parametric PLIF **(CPLIF)** neuron, thereby eliminating spike-driven temporal dynamics among inter-module.
> - In both **SBE** and **FBE** (branch-specific spiking encoders), all LIF neurons are replaced by standard ReLU activations.
> - The spiking channel self-attention **(SCSA)** module is replaced with a standard channel-wise self-attention, using typical QKV projections and softmax operations.
> - The spiking multi-scale convolution **(SMSC)** is converted into a  ANN conventions. All spiking neuron-based pre-processing blocks (e.g., $LIF$-$Conv$-$BN$) are similarly replaced by $Conv$-$BN$-$ReLU$.
> - In the spiking gated channel mixer (**SGCM** ) module,  spiking neuron-based pre-processing blocks (e.g., $LIF$-$Conv$-$BN$ and $LIF$-$DWConv$-$BN$-$LIF$) are converted to standard ANN forms, such as $Conv$-$BN$-$ReLU$ and $DWConv$-$BN$-$ReLU$.
> - The membrane potential-aware token mixer (**MPTM**) module fuses dual-branch token representations without any LIF neuron, as temporal spiking behavior is no longer present.
>
> All modifications are carefully made to preserve the overall architecture, layer depth, and channel dimensions, ensuring that **SM-Former serves as a structurally aligned ANN counterpart**. This enables a fair assessment of whether spiking computations offer tangible benefits over ANN counterparts.
>
> **We will add this clarification and include a more detailed description of SM-Former in the final version of the paper.**
>
> **2. Which part of the proposed framework contribute most to capture the long-range dependencies?**
>
> Thank you for the insightful question. The membrane potential-aware token mixer **(MPTM)** plays the most crucial role in modeling long-range dependencies within our framework.
>
> As detailed in Section 3.3.4 on Page 6 and illustrated in Fig. 2(f) on Page 4, MPTM enables global context modeling through a **three-stage** process: core representation, refinement and fusion, and output generation.
>
> In the **core** stage, global average pooling (GAP) condenses token-wise information into compact **global summaries**, allowing the model to aggregate contextual information across the entire sequence.
>
> In the **refinement and fusion** stage, these summaries are proportionally broadcast back to the sequence and used to modulate local token features via spiking gating and residual fusion, thereby **injecting long-range contextual cues into each token.**
>
> Finally, in the **output generation** stage, the globally modulated features are concatenated with compressed representations obtained via max pooling, enhancing representation diversity and supporting **context-aware encoding**.
>
> Through this **core → refine → fuse** process, MPTM facilitates global information integration and cross-branch interaction, making it a key contributor to the model’s capacity for capturing long-range dependencies essential for robust auditory attention decoding.
>
> We will highlight this role more clearly in the final version of the paper.
>
> **3. What might be the reason behind the narrowed advantage over prior non-SNN methods (e.g., DBPNet) in the KUL benchmark?**
>
> Thank you for this thoughtful question. While the performance margin of S²M-Former over prior ANN-based methods (e.g., DBPNet, M-DBPNet) appears narrower on the KUL benchmark, we ponder and believe this may stem from the **specific data characteristics** of KUL, rather than an inherent limitation of our model.
>
> KUL consists of only **8 long-duration trials (360s each)** under a **single, fixed auditory scene**, resulting in low trial diversity and highly repetitive EEG patterns. This **homogeneity** may limit the temporal variability that **spike-driven models are designed to exploit via their intrinsic temporal dynamics**. Moreover, the **within-trial setting**, where both training, validation and testing samples are drawn from the same trials, further reduces variability and encourages models to recognize trial-specific EEG patterns. Consequently, we believe the performance differences between models tend to narrow under this setting, not because S²M-Former lacks representational power compared to ANN-based methods, but rather due to the **low-complexity and low-variability** nature of the evaluation setup.
>
> Table 1: Within-trial results under KUL dataset (In all tables, **bold** indicates the best performance, and *italic* denotes the second-best.)
>
> | Models     | 2s               | 1s               | 0.1s             |
> | ---------- | ---------------- | ---------------- | ---------------- |
> | DBPNet     | 93.66 ± 7.88     | **93.25 ± 7.33** | *85.70 ± 9.75*   |
> | M-DBPNet   | **93.75 ± 6.34** | *93.19 ± 7.28*   | **86.16 ± 9.94** |
> | S²M-Former | *93.71 ± 8.14*   | 92.27 ± 8.66     | 83.39 ± 12.80    |
>
> However, the strengths of S²M-Former emerge more clearly in **cross-trial** and **leave-one-subject-out** evaluations under the KUL benchmark, where it matches or outperforms DBPNet and M-DBPNet in most cases (Table 2 and Table 3), demonstrating superior **generalization** under more challenging and realistic settings.
>
> Table 2: Cross-trial results under the KUL dataset
>
> | Models     | 2s                | 1s                | 0.1s              |
> | ---------- | ----------------- | ----------------- | ----------------- |
> | DBPNet     | *72.95 ± 24.36*   | *70.89 ± 25.01*   | 65.67 ± 21.84     |
> | M-DBPNet   | **74.27 ± 21.37** | 70.64 ± 23.82     | **66.97 ± 21.87** |
> | S²M-Former | 72.39 ± 25.21     | **71.22 ± 25.97** | *66.49 ± 21.03*   |
>
> Table 3: Leave-one-subject-out results under the KUL dataset
>
> | Models     | 2s                | 1s                |
> | ---------- | ----------------- | ----------------- |
> | DBPNet     | *74.82 ± 13.40*   | *71.77 ± 14.07*   |
> | M-DBPNet   | 72.83 ± 12.49     | 71.72 ± 13.35     |
> | S²M-Former | **75.75 ± 13.43** | **74.37 ± 12.57** |
>
> Furthermore, across more complex benchmarks such as **DTU (with more trials and shorter durations, specifically 60 trials × 50s )** and **AV-GC-AAD (audiovisual distractions with multi-condition)**, **S²M-Former** consistently outperforms prior models, confirming its effectiveness and **robustness across diverse AAD conditions**.
>
> We will clarify these points in the final version of the paper.

---

> > ### Comment · Reviewer_aRKz · 2025-08-02
> >
> > Thanks for the clarifications. The authors have solved most of my confusion and questions. The proposed method is overall effective and outperforms prior works, therefore, I will maintain a positive score.

---

### Official Review · Reviewer_qaAt · 2025-06-27

**Clarity:** 2
**Significance:** 3
**Originality:** 3
**Rating:** 4
**Confidence:** 3

**Summary:**

This paper introduces $S^2M-Former$ , a dual-branch, spike-driven architecture that learns complementary spatial (CSP) and frequency-spectral (DE) EEG features through a mirrored stack of spike-friendly modules. The method was tested on three different datasets and achieved state-of-the-art (SOTA) classification results.

**Questions:**

1) In Section 3.3.2 (SMSC), how is the channel shuffle operation implemented in practice? Is there any difference in the usage during training and testing? It would be interesting to see a performance comparison with and without the channel shuffle.
2) Is the CSP (Common Spatial Pattern) process applied before or after dataset partitioning.
3) As the number of channels and decision window length vary, the parameter count will also change. What specific configuration was used to determine the reported 0.06M parameters?
4) Did the authors use sliding window processing during the training phase? Since all compared methods adopt sliding window processing, was the same strategy consistently applied across all methods for fair comparison?

**Ethical Concerns:**

["NO or VERY MINOR ethics concerns only"]

**Final Justification:**

The paper proposes a novel spike-driven symmetric architecture for auditory attention decoding. While achieving state-of-the-art performance across three benchmark datasets, the method maintains minimal computational cost, reporting only 0.06M parameters. Besides, The authors have addressed most of my concerns and answered the key questions during rebuttal. But the method and motivation in the current draft are not clearly written. I suggest the authors consider rephrasing this part in the final version to improve clarity. Therefore, I have increased my socre from 3 to 4.

**Limitations:**

YES.

**Paper Formatting Concerns:**

NO or VERY MINOR ethics concerns only

**Quality:**

3

**Strengths And Weaknesses:**

# Strength
+ The paper proposes a novel spike-driven symmetric architecture for auditory attention decoding.
+ While achieving state-of-the-art performance across three benchmark datasets, the method maintains minimal computational cost, reporting only 0.06M parameters.
+ The paper provides comprehensive and extensive experiments, including evaluations under within-trial, cross-trial, and cross-subject settings, as well as detailed ablation studies.
# Weakness
+ Some definitions are unclear or missing. For example, DWConv and PWConv.
+ Although the authors emphasize the importance of reducing computational cost, it's unclear why this method matters.

---

> ### Author Rebuttal · Authors · 2025-07-28
>
> Dear Reviewer qaAt,
>
> Thank you for the detailed comments and insightful suggestions. We genuinely appreciate your insights and will address your concerns point by point below.
>
> ### **[Weakness]:**
>
> **1. Some definitions are unclear or missing. For example, DWConv and PWConv.**
>
> Thank you for pointing this out. We apologize for the omission and will clarify the definitions in the final version. Specifically:
>
> - DWConv refers to depthwise convolution, which handles each channel separately with its unique filter.
> - PWConv refers to pointwise convolution, i.e., a 1×1 convolution used to project features across channels.
>
> We will revise the descriptions accordingly, such as:
>
> Line 188 of page 5: "..., followed by point-wise convolution **(PWConv)** and batch normalization."
>
> Line 190 of page 5: "..., and each part is passed through a **depthwise convolution** (DWConv1d$_{k_i}$) with ..."
>
> **2. Although the authors emphasize the importance of reducing computational cost, it's unclear why this method matters.**
>
> Thank you for the thoughtful question.  AAD is increasingly being integrated into low-power neuro-steered devices, such as **smart hearing aids [1], wearable EEG headsets [2], and earable BCIs [3]**. These systems are inherently battery-powered and face strict constraints on energy, latency and processing efficiency, especially under continuous operations in real-world environments.
>
> For instance, a neuro-steered hearing aid that continuously tracks a user's attentional focus in noisy environments requires energy-efficient, lightweight models for long-term usability. In such scenarios,  **S²M-Former** is particularly advantageous, as it achieves higher decoding accuracy while significantly reducing **parameter count and theoretical energy consumption** compared to recent ANN-based dual-branch models (e.g., DBPNet, M-DBPNet), as shown in Table 5.
>
> This efficiency stems not only from a **unified and symmetric architecture** that processes lightweight 1D token sequences without requiring complex branch-specific designs (e.g., deep 3D CNN blocks for topographic features), but also importantly from the **fully spike-driven, brain-inspired paradigm**, where internal computations are triggered **only when spikes (i.e., 1) are fired** via accumulate (AC) operations, unlike multiply-accumulate (MAC) operations requiring costly floating-point computations. This **asynchronous, event-driven computation** avoids unnecessary activity, making S²M-Former particularly well-suited for long-term monitoring, enabling low-power deployment in wearable AAD systems.
>
> We will clarify and emphasize this point in the final version of the paper.
>
> [1] Van Eyndhoven S, et al. EEG-informed attended speaker extraction from recorded speech mixtures with application in neuro-steered hearing prostheses [J]. IEEE Transactions on Biomedical Engineering, 2016.
>
> [2] Mirkovic B, et al. Decoding the attended speech stream with multi-channel EEG: implications for online, daily-life applications [J]. Journal of Neural Engineering, 2015.
>
> [3] Fiedler L, et al. Single-channel in-ear-EEG detects the focus of auditory attention to concurrent tone streams and mixed speech [J]. Journal of Neural Engineering, 2017.
>
> ### **[Questions]:**
>
> **1. How is the channel shuffle operation implemented in practice? A performance comparison with and without the channel shuffle.**
>
> We adopt a group-wise **channel shuffle operation** (as discussed in Section 3.3.2 on Page 5) to enhance inter-group information exchange after multiple depth-wise convolutions. Given an input tensor of shape `[Ts, B, N, D]`  (as in Eq. (11) of the paper), where `D` is divisible by the number of groups `g`, we perform the following steps: 1. **Reshape** to `[Ts, B, N, g, D/g]`; 2. **Transpose** the group and channel dimensions; 3. **Flatten** back to the original shape `[Ts, B, N, D]`.
>
> This process is implemented as:
>
> ```
> def channel_shuffle(self, x, groups):
>     """
>     Ts: time steps, B: batch size, D: number of channels, N: number of tokens.
>     """
>     Ts, B, N, D = x.size()
>     channels_per_group = D // groups
>     x = x.view(Ts, B, N, groups, channels_per_group) # (Ts, B, N, g, D/g)
>     x = x.transpose(3, 4).contiguous() # (Ts, B, N, D/g, g)
>     x = x.view(Ts, B, N, D)            # (Ts, B, N, D)
>     return x
> ```
>
> Importantly, the shuffle operation is purely tensor rearrangement and is **identical during training and testing**, introducing no learnable parameters.
>
> To assess its contribution, we are conducting ablation experiments by removing the **channel shuffle (CS)**.  Results across three datasets are summarized below:
>
> Table 1: DTU dataset comparison
>
> | Model          | Within-trial (2s) | Within-trial (1s) | Within-trial (0.1s) | Cross-trial (2s) | Cross-trial (1s) | Cross-trial (0.1s) |
> | -------------- | ----------------- | ----------------- | ------------------- | ---------------- | ---------------- | ------------------ |
> | **S²M-Former** | 85.28 ± 6.01      | **82.87 ± 6.92**  | **75.84 ± 5.46**    | **76.74 ± 9.96** | **75.75 ± 9.96** | **70.36 ± 7.31**   |
> | **w/o CS**     | **85.39 ± 6.27**  | 82.31 ± 6.60      | 75.83 ± 5.85        | 75.70 ± 10.38    | 75.28 ± 10.24    | 69.69 ± 7.58       |
>
> Table 2:  KUL dataset comparison
>
> | Model          | Within-trial (2s) | Within-trial (1s) | Within-trial (0.1s) | Cross-trial (2s)  | Cross-trial (1s)  | Cross-trial (0.1s) |
> | -------------- | ----------------- | ----------------- | ------------------- | ----------------- | ----------------- | ------------------ |
> | **S²M-Former** | **93.71 ± 8.14**  | **92.27 ± 8.66**  | **83.39 ± 12.80**   | **72.39 ± 25.21** | **71.22 ± 25.97** | **66.49 ± 21.03**  |
> | **w/o CS**     | 93.25 ± 8.81      | 92.01 ± 9.52      | 82.46 ± 13.36       | 71.51 ± 25.93     | 70.43 ± 25.60     | 66.29 ± 20.77      |
>
> Table 3:  AV-GC dataset comparison
>
> | Model          | Within-trial (2s) | Within-trial (1s) | Within-trial (0.1s) | Cross-trial (2s)  | Cross-trial (1s)  | Cross-trial (0.1s) |
> | -------------- | ----------------- | ----------------- | ------------------- | ----------------- | ----------------- | ------------------ |
> | **S²M-Former** | **91.83 ± 6.66**  | **89.24 ± 7.59**  | **74.42 ± 7.79**    | 70.64 ± 18.65     | **65.77 ± 15.58** | **65.49 ± 13.74**  |
> | **w/o CS**     | 91.18 ± 6.30      | 88.59 ± 7.35      | 74.26 ± 7.48        | **70.65 ± 18.47** | 65.60 ± 13.83     | 65.42 ± 15.64      |
>
> As shown in Table 1-3, **removing channel shuffle leads to consistent drops** in accuracy under most settings. This supports its effectiveness in enhancing inter-group diversity. We will include full results and discussion for all decision windows and datasets in the final version.
>
> **2. Is the CSP process applied before or after dataset partitioning.**
>
> We appreciate your attention to this detail.  CSP was applied **AFTER dataset partitioning.** As stated on **Page 7, Lines 254–255,**
>
> >"**We ensure no overlap between sets after splitting and perform separate feature extraction within each set to avoid any potential information leakage.**"
>
> This process strictly avoids any potential information leakage from test data into the CSP feature extraction.
>
> **3. How was the 0.06M parameter count determined？**
>
> Thank you for your question. We would like to clarify that the reported parameter count of **0.06M remains constant across all decision window lengths and datasets.**
>
> As detailed in **Eq. (2)–Eq. (18)**, all components in S²M-Former, including branch-specific spiking encoders (SBE, FBE) and S²M Module (SCSA, SMSC, SGCM and MPTM), **contain learnable operations only through channel-wise projections (D)**.  Each component is accompanied by explicit tensor shape definitions (e.g.,  $\mathbb{R}^{T_S \times N \times D}$), clearly illustrating that no learnable transformations along the token dimension (N), which corresponds to the decision window length in the spatial branch and the topographic map resolution in the frequency branch.  As such, varying the number of tokens does not affect the model’s parameter count.
>
> For clarity, operations such as token dimension concatenation (Page 5, Lines 195–196), global average pooling (Eq. (15)), and max pooling (Eq. (18)) do not introduce additional parameters, and thus do not impact the overall parameter count. Moreover, as specified on Page 14, Line 578,
>
> >**"The hidden dimension D in S²M-Former is set to 8."**
>
> We set the hidden dimension to $D=8$ for all experiments, and all three datasets (KUL, DTU, AV-GC-AAD) employ a uniform 64-channel EEG configuration, ensuring consistent input dimensionality. Therefore, the reported **0.06M parameters** reflect a unified model configuration used across all datasets and settings. We will clarify this point more explicitly in the final version of the paper.
>
> **4. Did the authors use sliding window processing during training? Was the same strategy used across all methods for fair comparison?**
>
> Thank you for the insightful question. **YES,** we **adopt the same sliding window strategy as prior works** to ensure fair comparison. As stated on Page 6, Lines 239–240,
>
> >**"The pre-processed EEG data are segmented using a sliding window with 50% overlap, where the window step size is half the time window length."**
>
> The **unified** data preprocessing pipeline is consistently applied across all datasets and all methods (including baselines and our model). To ensure **fair benchmarking**, we reproduce and evaluate all compared methods under the same implementation, including window length, step size, and overlap ratio. This setup follows **standard practice adopted in prior AAD works** (e.g., M-DBPNet, DBPNet, DARNet), ensuring that observed performance gains arise from model innovations rather than differences in data segmentation or preprocessing.
>
> To avoid ambiguity and facilitate reproducibility, we will strengthen this clarification in the final version and release our code upon acceptance for transparent validation.

---

> > ### Author Response · Authors · 2025-08-04
> >
> > Dear Reviewer qaAt,
> > ﻿
> >
> > We truly appreciate your thoughtful comments, which have helped us strengthen both the presentation and analysis of our work. We sincerely hope that our clarifications and the additional results provided have effectively addressed your concerns. To provide greater clarity on key aspects such as the CSP pipeline, parameter count consistency, and sliding window preprocessing, we have expanded on these points in our rebuttal, including additional ablation studies (e.g., the impact of channel shuffle).
> > ﻿
> >
> > Should there be any remaining questions or concerns, we would be more than happy to provide further details or engage in further discussion.
> > ﻿
> >
> > Thank you again for your constructive feedback and the time you dedicated to reviewing our submission.

---

> > ### Comment · Reviewer_qaAt · 2025-08-04
> >
> > Thank you for the clarifications.
> >
> > Regarding Weakness 2, my original intention was not to question the importance of computational efficiency per se, but rather to ask for a clearer explanation of why the proposed method is effective.
> >
> > For Question 1, could you provide more insight into the underlying principle of the channel shuffle operation? I would be willing to increase my score if this point is addressed thoroughly.
> >
> > For Question 3, please note that models such as M-DBPNet, DBPNet, and DARNet have parameter counts that vary with the decision window length, whereas your model’s parameter count remains constant. I recommend that you make this clear in the paper and, for fair comparison, ensure that all models are compared under the same decision window length.

---

> > > ### Author Response · Authors · 2025-08-05
> > >
> > > Dear Reviewer qaAt,
> > >
> > > Thank you again for your thoughtful follow-up questions. We appreciate the opportunity to provide further clarification.
> > >
> > > ---
> > >
> > > **Q1: Why the proposed method is effective .**
> > >
> > > Thank you for the clarification. The effectiveness of S²M-Former stems from both its **architectural design** and its **spike-driven computational paradigm**, which together enable accurate and efficient AAD decoding.
> > >
> > > **1. Symmetric Complementary Architecture**:
> > >
> > > S²M-Former adopts a **unified and symmetric spike-driven architecture**, where both branches operate on lightweight 1D token sequences. Each branch employs efficient intra-branch modules (SCSA and SMSC) for **hierarchical feature learning**, while shared inter-branch fusion modules (SGCM and MPTM) enable **synergistic integration** of spatial and frequency features. This design provides strong representational capacity with minimal overhead and facilitates **complementary learning** between spatial and frequency cues essential for auditory attention decoding.
> > >
> > > **2. Spike-driven Computation**
> > >
> > > The entire model operates under a fully spike-driven paradigm, with **intra-module computation** handled by LIF neurons and **inter-module communication** mediated by CPLIF neurons. This setup enables **asynchronous, event-driven processing**, where operations are triggered only when spikes occur, achieving both **temporal dynamic modeling** and **energy efficiency**.
> > >
> > > Extensive experiments show that S²M-Former achieves both high efficiency and strong decoding performance, validating that its effectiveness stems from the combination of a symmetric architecture and spike-driven computation.
> > >
> > > We will revise the final paper to highlight this explanation more explicitly.
> > >
> > > ---
> > >
> > > **Q2: The underlying principle of the channel shuffle operation.**
> > >
> > > Certainly. To compensate for the lack of inter-channel communication in depthwise convolutions, we introduce a **channel shuffle operation** [1] in the SMSC module (Fig. 2(d), Page 4) to **enable cross-channel information flow** and enhance representational capacity.
> > >
> > > In SMSC, the expanded channels are split into three branches, each processed by a depthwise convolution (**DWConv**) with kernel sizes 1, 3, or 5 to enable **lightweight multi-scale feature extraction**. However, depthwise convolutions apply **independent filters to each channel**, unlike standard convolutions which aggregate information across channels. This reduces computational cost but may hinder expressive capacity due to **the absence of cross-channel interaction**.
> > >
> > > To mitigate this limitation while preserving efficiency, we introduce a channel shuffle operation after the DWConv branches. Specifically, the channels are first divided into groups, and then their positions are **reassigned across these groups**. This lightweight, parameter-free operation encourages **feature mixing across the outputs of different DWConv kernels**, enabling richer representation learning in subsequent inter-branch fusion.
> > >
> > > For implementation details, please refer to our earlier response.
> > >
> > > As demonstrated in the ablation results (Tables 1–3), removing channel shuffle consistently degrades performance, confirming its essential role in preserving representation quality under efficient design constraints.
> > >
> > > **We will highlight this principle more explicitly in the final version.**
> > >
> > > [1] Zhang X, et al. Shufflenet: An extremely efficient convolutional neural network for mobile devices[C] // CVPR. 2018.
> > >
> > > ---
> > >
> > > **Q3:  Parameter Count Consistency and Fair Evaluation Settings**
> > >
> > > Thank you for this valuable recommendation. We fully agree that parameter count consistency and fair comparison are critical.
> > >
> > > To clarify, among the referenced models, **M-DBPNet** exhibits window-dependent parameter counts due to the inclusion of the Mamba module, as explicitly reported in Table 1 and Table 2 (*1.32M / 1.00M / 0.88M*).
> > >
> > > In contrast, **DBPNet** and **DARNet** maintain fixed parameter counts regardless of window size. We carefully verified all model configurations by reproducing each method using their publicly released code, and cross-checked our implementation against logs, checkpoints, and their paper descriptions to ensure consistency.
> > >
> > > We sincerely appreciate your suggestion and will revise the final version of the paper to **more clearly highlight** that:
> > >
> > > - Emphasize that **S²M-Former's parameter count remains constant across different decision window lengths**;
> > > - Guarantee that **all models were evaluated under consistent window lengths** (0.1s, 1s, 2s);
> > > - Double check and clarify that **parameter counts for DBPNet, M-DBPNet, and DARNet were fairly accounted for in our comparison**.
> > >
> > > ---
> > >
> > > We hope these clarifications address your concerns. Thank you again for your valuable feedback.

---

> > > > ### Comment · Reviewer_qaAt · 2025-08-06
> > > >
> > > > Thank you for the clarifications. I will increase my score from 3 to 4.

---

### Official Review · Reviewer_Ypre · 2025-07-01

**Clarity:** 2
**Significance:** 2
**Originality:** 2
**Rating:** 3
**Confidence:** 3

**Summary:**

This paper introduces S2M-Former, a spiking neural network for auditory attention detection (AAD) that combines energy efficiency with effective fusion of complementary EEG features. The model uses a symmetric architecture with spatial and frequency branches, along with lightweight 1D token sequences, achieving competitive accuracy while significantly reducing power consumption and model size.

**Questions:**

- The paper emphasizes energy efficiency as a key contribution of the proposed SNN-based model. However, it is not entirely clear why energy consumption is a critical factor in the context of auditory attention detection. Could the authors elaborate on the specific application scenarios where this improvement would be particularly impactful?

- While the results show that the SNN model achieves performance comparable to or slightly better than ANN-based approaches, it’s unclear what exactly drives this effectiveness — is it the spiking architecture itself, or simply the architectural design choices inspired by SNNs? Would an equivalent ANN with similar inductive biases potentially perform even better?

**Ethical Concerns:**

["NO or VERY MINOR ethics concerns only"]

**Final Justification:**

The rebuttal partially clarifies the methodological aspects, but the core issues remain. First, the application of SNNs to EEG processing is well-established in the literature [1-3], and the proposed method does not demonstrate significant novelty. Second, the scope is limited to Auditory Attention Detection, which restricts the broader impact of this work.

Furthermore, the power consumption concern, while acknowledged, is not a primary bottleneck in EEG processing. The field already has numerous lightweight models (e.g., ShallowConvNet, EEGNet) with minimal power requirements. Therefore, the claimed advantages in this aspect are not compelling enough to justify the incremental contribution.

Given these considerations, I maintain my original evaluation.

[1] Kumar, Neelesh, et al. "Decoding eeg with spiking neural networks on neuromorphic hardware." Transactions on Machine Learning Research (2022).

[2] Antelis, Javier M., and Luis E. Falcón. "Spiking neural networks applied to the classification of motor tasks in EEG signals." Neural networks 122 (2020): 130-143.

[3] Gong, Peiliang, et al. "A spiking neural network with adaptive graph convolution and LSTM for EEG-based brain-computer interfaces." IEEE Transactions on Neural Systems and Rehabilitation Engineering 31 (2023): 1440-1450.

**Limitations:**

yes

**Quality:**

3

**Strengths And Weaknesses:**

**Strengths**

- Introducing spiking neural networks (SNNs) into EEG-based auditory attention detection (AAD) is a novel and promising direction, especially given the growing interest in energy-efficient brain-computer interfaces. This work explores how such models can be adapted to this specific decoding task.

- The proposed symmetric architecture with hierarchical integration of spatial and frequency features seems conceptually reasonable for capturing contextual EEG patterns, aligning with how auditory processing unfolds over time and across brain regions.

- The experimental evaluation is reasonably comprehensive, involving three publicly available AAD datasets (KUL, DTU, AV-GC-AAD) across multiple settings (within-trial, cross-trial, cross-subject), which adds credibility to the reported results.


**Weaknesses**

- While the application of SNNs to AAD is interesting, the methodological novelty appears limited. Using spiking networks in EEG decoding is not entirely new, and the main contribution seems to lie in adapting and optimizing existing architectures rather than introducing fundamentally new concepts.

- The hybrid design that combines elements from both SNNs and ANNs raises questions about its practical deployment on neuromorphic hardware. It's unclear whether such a mixed framework can truly benefit from the expected energy efficiency gains in real-world scenarios.

- The performance improvements over existing methods are modest at best. Moreover, while the paper highlights energy efficiency as a key advantage, Table 5 shows only moderate reductions in theoretical energy consumption compared to conventional ANN models — suggesting the gains may not be as substantial as claimed.

- The writing and presentation of the method could be improved. The model description heavily relies on equations with insufficient explanation of key symbols and components, making it difficult to fully grasp the design choices and their implications.

---

> ### Author Rebuttal · Authors · 2025-07-28
>
> Dear Reviewer Ypre,
>
> Thank you for your detailed comments. We genuinely appreciate your insights and will address your concerns point by point below.
>
> ### **[Weakness]:**
>
> **1. While the application of SNNs to AAD is interesting, the methodological novelty appears limited ...**
>
> We believe our work introduces **S²M-Former**, a fundamentally **unified, symmetric spike-driven framework** specifically tailored for dual-feature EEG-based AAD, advancing beyond prior SNN and ANN approaches.
>
> Unlike recent methods that **treat branches independently and rely on weak or late-stage feature fusion**, S²M-Former innovatively leverages intra-branch modules for **hierarchical feature learning** and the shared inter-branch fusion module for **synergistic integration**, ensuring both accuracy and efficiency.
>
> As also recognized by Reviewer aRKz, our S²M-Former is the first spiking framework to hierarchically integrate spatial-temporal and topographic EEG features, rather than relying on simple feature concatenation.
>
> Importantly, the inherently sparse and binary nature of spike representations poses a key challenge that has limited recent SNNs from matching ANN performance on AAD tasks (see *Related Work*). Our model outperforms recent SOTA ANN counterparts (e.g., DBPNet, M-DBPNet) on multiple AAD benchmarks, while using 14.7× fewer parameters and achieving 5.8× lower theoretical energy consumption. This overcomes **the long-standing challenge that existing SNNs trade accuracy for efficiency** and opens a new direction for high-performance, low-power AAD.
>
> In this sense, our work contributes not only an effective AAD solution, but also a novel demonstration that **spiking neural architectures can scale to dual-feature, high-accuracy EEG decoding**, moving beyond the limitations of previous SNN applications in this domain.
>
> **2.  The hybrid design that combines elements from both SNNs and ANNs ...**
>
> We would like to clarify a potential misunderstanding regarding "hybrid design". S²M-Former is implemented entirely under a purely spike-driven computation paradigm. Specifically, all internal layers perform with **intra-module computation** via LIF neurons, while **inter-module communication** is handled by CPLIF neurons (explicitly described in **Section 3.1, line 125-127** on Page 3).
>
> Components such as convolution and linear layers are fundamental to representation learning and widely used in both ANN and SNN models, their presence does not imply a hybrid architecture [1]. The **key factor** that determines whether a model is hybrid ANN-SNN or pure SNN lies in whether its internal computation and communication are truly spike-driven [2]. As such, S²M-Former performs internal computation **only when spikes are fired**, relying on accumulate operations rather than floating-point based multiply-accumulate operations.
>
> This enables full compatibility with neuromorphic platforms such as Intel Loihi or Tianjic, which support event-driven execution. We have already converted the trained weights of S²M-Former into a format compatible with Intel Loihi-2 [3] by leveraging LAVA framework, and we are actively pursuing access to neuromorphic hardware for evaluation. We will incorporate this clarification in the final version of the paper.
>
> [1] Eshraghian J K, et al. Training spiking neural networks using lessons from deep learning [J]. Proceedings of the IEEE, 2023.
>
> [2] Kudithipudi D, et al. Neuromorphic computing at scale [J]. Nature, 2025.
>
> [3] Orchard G, et al. Efficient neuromorphic signal processing with loihi 2 [C]. 2021 IEEE Workshop on Signal Processing Systems (SiPS), 2021.
>
> **3. Table 5 shows only moderate reductions in theoretical energy consumption ...**
>
> Thank you for this valuable observation. We would like to clarify the context behind our energy-efficiency claims.
>
> As shown in **Table 2, 3, Fig. 4 and 5**, dual-branch models consistently outperform single-branch ones, supporting the hypothesis (Page 2, Lines 38–55) that complementary learning between spatial-temporal and topographic EEG features enhances decoding performance. However, most existing dual-branch ANN-based AAD models (e.g., DBPNet, M-DBPNet) rely on **isolated** feature learning paradigms, with each branch built upon distinct architectures. This separation not only limits the ability to learn cross-domain dependencies but also requires **customized feature extractors** (e.g., deep 3D CNN blocks), leading to substantially higher parameter counts, computational cost, and energy consumption.
>
> In contrast, S²M-Former adopts a **unified, symmetric spike-driven architecture**, where both branches operate on lightweight 1D token sequences, leveraging efficient intra-branch modules for feature learning and the shared inter-branch fusion modules for synergistic integration. This design supports strong representational capacity with minimal overhead, achieving up to **14.7× fewer parameters** and **5.8× lower theoretical energy** while outperforming DBPNet and M-DBPNet (Table 5).
>
> Therefore, we believe **the observed energy reductions** are **substantial and meaningful**, especially when **contextualized against high-performing dual-branch models**. S²M-Former offers a favorable performance-energy trade-off that is critical for real-world, low-power AAD deployment.
>
> **4. The writing and presentation of the method could be improved.**
>
> Thank you for the helpful feedback. While we have rigorously defined all variables and detailed equations (Eq.(2)–Eq.(18)), we acknowledge that parts of the model description may still lack intuitive explanation. To address this, we will refine the text and more clearly align symbolic expressions with Fig. 2 to enhance presentation in the final version.
>
> ### **[Questions]:**
>
> **1. Why energy consumption is a critical factor in the context of AAD ?**
>
> Thank you for raising this important point. AAD systems are widely recognized as crucial components of neuro-steered hearing technologies, such as **smart hearing aids [1], wearable EEG headsets [2], and earable BCIs [3]**, which are typically battery-powered and constrained by power, latency and computational resources, particularly in dynamic, real-world conditions.
>
> In this context, energy-efficient AAD models are vital to ensure **long battery life, user comfort, and practical deployment.** As highlighted in several studies [1–3], current AAD systems are often too resource-intensive for such applications. Our S²M-Former addresses this with a lightweight spike-driven architecture that significantly reduces computational cost.
>
> For instance, a neuro-steered hearing aid that continuously decodes a user's attentional focus to selectively amplify the attended speaker in noisy environments. Such a device must operate reliably for hours on limited hardware and battery. In such scenarios, the lightweight, spike-driven design of S²M-Former is especially impactful, not only due to its low computational complexity, but also for its event-driven nature, **where computations are triggered only by discrete spikes.** **This asynchronous paradigm** better aligns with EEG dynamics in long-term monitoring, minimizing power consumption and facilitating efficient deployment in wearable AAD systems.
>
> This clarification will be included in the final paper.
>
> [1] Van Eyndhoven S, et al. EEG-informed attended speaker extraction from recorded speech mixtures with application in neuro-steered hearing prostheses [J]. IEEE Transactions on Biomedical Engineering, 2016.
>
> [2] Mirkovic B, et al. Decoding the attended speech stream with multi-channel EEG: implications for online, daily-life applications [J]. Journal of Neural Engineering, 2015.
>
> [3] Fiedler L, et al. Single-channel in-ear-EEG detects the focus of auditory attention to concurrent tone streams and mixed speech [J]. Journal of Neural Engineering, 2017.
>
> **2. Would an equivalent ANN with similar inductive biases potentially perform even better ?**
>
> Actually, we **have already examined this** in our main paper by designing a matched ANN counterpart, termed **SM-Former**, where all spike-driven modules in S²M-Former are replaced with standard ANN components while preserving the overall architecture and parameter count. (Please refer to our response to **Reviewer aRKz – Q1** for the detailed SNN-to-ANN conversion procedure.)
>
> As shown in **Tables 4, 8, and 9** of the main paper (reproduced below for clarity), S²M-Former consistently outperforms SM-Former across most datasets and evaluation settings, demonstrating that the performance gain stems not just from architectural design, but from the **spike-driven computation paradigm itself**.
>
> Table 1: Comparison with SM-Former on Three AAD Datasets.
>
> | Dataset | Model      | Within-trial (2s) | Within-trial (1s) | Cross-trial (2s)  | Cross-trial (1s)  |
> | ------- | ---------- | ----------------- | ----------------- | ----------------- | ----------------- |
> | DTU     | S²M-Former | **85.28 ± 6.01**  | **82.87 ± 6.92**  | **76.74 ± 9.96**  | **75.75 ± 9.96**  |
> |         | SM-Former  | 80.94 ± 5.66      | 80.30 ± 6.77      | 73.49 ± 8.09      | 73.48 ± 7.98      |
> | KUL     | S²M-Former | **93.71 ± 8.14**  | **92.27 ± 8.66**  | **72.39 ± 25.21** | **71.22 ± 25.97** |
> |         | SM-Former  | 91.38 ± 9.71      | 90.65 ± 10.80     | 69.46 ± 23.75     | 66.69 ± 23.52     |
> | AV-GC   | S²M-Former | **91.83 ± 6.66**  | **89.24 ± 7.59**  | **70.64 ± 18.65** | 65.77 ± 15.58     |
> |         | SM-Former  | 90.66 ± 6.84      | 88.07 ± 7.10      | 68.83 ± 17.50     | **66.52 ± 13.75** |
>
> These results clearly show that S²M-Former not only retains the inductive biases of its ANN counterpart, but also benefits from the temporal dynamics, event-driven, and biological plausibility offered by spiking computation. This validates our core motivation: **spike-based processing can achieve superior performance while being more energy-efficient.**
>
> We will make this point more prominent and explicit in the final version.

---

> > ### Comment · Reviewer_Ypre · 2025-08-03
> >
> > Thank you for your thoughtful rebuttal. While the authors have addressed some of my concerns, I maintain my original score for the following reasons:
> >
> > The rebuttal partially clarifies the methodological aspects, but the core issues remain. First, the application of SNNs to EEG processing is well-established in the literature [1-3], and the proposed method does not demonstrate significant novelty. Second, the scope is limited to Auditory Attention Detection, which restricts the broader impact of this work.
> >
> > Furthermore, the power consumption concern, while acknowledged, is not a primary bottleneck in EEG processing. The field already has numerous lightweight models (e.g., ShallowConvNet, EEGNet) with minimal power requirements. Therefore, the claimed advantages in this aspect are not compelling enough to justify the incremental contribution.
> >
> > Given these considerations, I maintain my original evaluation.
> >
> > [1] Kumar, Neelesh, et al. "Decoding eeg with spiking neural networks on neuromorphic hardware." Transactions on Machine Learning Research (2022).
> >
> > [2] Antelis, Javier M., and Luis E. Falcón. "Spiking neural networks applied to the classification of motor tasks in EEG signals." Neural networks 122 (2020): 130-143.
> >
> > [3] Gong, Peiliang, et al. "A spiking neural network with adaptive graph convolution and LSTM for EEG-based brain-computer interfaces." IEEE Transactions on Neural Systems and Rehabilitation Engineering 31 (2023): 1440-1450.

---

### Author Response · Authors · 2025-08-04

Dear Reviewers aRKz, qaAt, and Ypre,

We sincerely thank all three reviewers for your careful reading, valuable feedback, and constructive suggestions. Your comments have greatly helped us improve the clarity, rigor, and overall quality of our work. We deeply appreciate your time and efforts during the review process.

---

### Note · Authors · 2025-08-11

We thank all reviewers for their time and constructive feedback, and use this opportunity to summarize how concerns were addressed in the rebuttal.

For **Reviewer Ypre**, we:

- Clarified that **S²M-Former** is the *first unified, symmetric spike-driven framework* to hierarchically fuse spatial and frequency EEG features, overcoming the accuracy–efficiency trade-off of prior SNNs.
- Emphasized the *purely spike-driven* nature of all modules, ensuring neuromorphic compatibility and event-driven execution.
- Established energy advantages as *substantial* relative to dual-branch baselines (DBPNet, M-DBPNet).
- Verified that gains stem from the spike-driven paradigm *itself*, via matched ANN counterpart (SM-Former) experiments.
- Justified energy efficiency as *central* to real-world AAD deployment by highlighting concrete application scenarios.

While the score remained unchanged, **major concerns were addressed on record**.

For **Reviewer qaAt**, we:

- Explained the channel shuffle *principle* in SMSC for inter-group feature mixing, supported by *ablation results*.
- Clarified that S²M-Former’s parameter count is *window-length invariant*, and that all baselines were evaluated under identical settings.
- Confirmed *consistent* EEG preprocessing (CSP *after* data split, *same* sliding window strategy) across all models.

These clarifications were acknowledged and the reviewer **raised their score**.

For **Reviewer aRKz**, we:

- Analyzed the narrower KUL within-trial gains caused by dataset *homogeneity*, and emphasized stronger results in cross-trial and LOSO settings.
- Provided a detailed *conversion description* of SM-Former to ensure a fair SNN–ANN comparison.
- Identified MPTM as the *main contributor* to long-range dependency modeling via its *core–refine–fuse* process.

The reviewer confirmed that concerns were resolved and maintained a **positive score**.

**In summary,** our rebuttal:

- Delivered evidence-backed clarifications and results addressing key concerns;
- Secured a *score increase* from Reviewer qaAt and *clear recognition* from Reviewer aRKz;
- Reinforced S²M-Former’s **novelty, robustness, and practical relevance**,  achieving SOTA AAD decoding with *fewer* parameters and *lower* theoretical energy than strong dual-branch ANN baselines.

We respectfully trust that the AC will consider the resolved concerns, improved reviewer assessments, and the broader impact of our work on energy-efficient, high-performance EEG-based AAD decoding.

---

### Decision · Program_Chairs · 2025-09-17

**Decision:**

Accept (poster)

**Comment:**

The paper presents S^2M-Former, a spiking symmetric mixing framework devised for auditory attention detection (AAD) using EEG signals. It adopts a symmetric architecture with spatial and frequency branches, along with lightweight 1D token sequences to replace conventional 3D operations. Unlike traditional methods that naively concatenate features, S^2M-Former uses hierarchical integration to fuse spatial-temporal EEG dynamics with brain topographical maps. The proposed approach obtains state-of-the-art classification performance across three publicly available AAD datasets while significantly reducing model size and energy consumption.

Strengths include:  spiking neural networks (SNNs) for EEG-based AAD, symmetric architecture, evaluation on three publicly available datasets, and state-of-the-art performance with reduced computational cost. Weaknesses include: moderate methodological novelty, modest and inconsistent performance gains over prior methods,   the need for elaborating on the model efficiency aspects.

The rebuttal process helped shed light on the concerns.